# GENERALIZATION OF GIBBS AND LANGEVIN MONTE CARLO ALGORITHMS IN THE INTERPOLATION REGIME

## ABSTRACT

The paper provides data-dependent bounds on the test error of the Gibbs algorithm in the overparameterized interpolation regime, where low training errors are also obtained for impossible data, such as random labels in classification. The bounds are stable under approximation with Langevin Monte Carlo algorithms. Experiments on the MNIST and CIFAR-10 datasets verify that the bounds yield nontrivial predictions on true labeled data and correctly upper bound the test error for random labels. Our method indicates that generalization in the low-temperature, interpolation regime is already signaled by small training errors in the more classical high temperature regime.

## 1 INTRODUCTION

Modern learning algorithms can achieve very small training errors on arbitrary data if the underlying hypothesis space is large enough. For reasonable data originating from real-world problems, the chosen hypotheses also tend to have small test errors, a fortunate circumstance, which has given great technological and economic thrust to deep learning. Unfortunately, the same algorithms also achieve very small training errors for data specifically designed to produce very large test errors, such as random labels in classification. Consequently, the hypothesis space and the training error do not suffice to predict the test error. The key to generalization must be more deeply buried in the data. While not so disquieting to practitioners, this mystery has troubled theoreticians for many years (Zhang et al., 2016; 2021), and it seems safe to say that the underlying mechanisms still have not been completely understood.

We are far from solving this riddle in generality, but for the Gibbs posterior we show how nontrivial bounds on the test error can be recovered from the training data. The Gibbs posterior assigns probabilities, which decrease exponentially with the training error of the hypotheses. The exponential decay parameter $\beta$ can be interpreted as an inverse temperature in an analogy to statistical physics. The Gibbs measure is a sufficient idealization to have tractable theoretical properties, but also the limiting distribution of several concrete stochastic algorithms, here summarized as Langevin Monte Carlo (LMC), including Stochastic Gradient Langevin Dynamics (SGLD), (Gelfand & Mitter, 1991; Welling & Teh, 2011), a popular modern learning algorithm.

When $\beta$ is large and the hypothesis space is rich, these algorithms can reproduce the dilemma described above by achieving very small training errors on data designed to have large test errors. Our paper addresses this *interpolation regime* of the Gibbs posterior and makes the following three contributions:

- We give high-probability data-dependent bounds on the test error, both for a hypothesis drawn from the Gibbs posterior and for the posterior mean, assuming that we can freely draw samples from it. These bounds holds for the entire range of temperatures.

- We weaken the above assumption by showing that the bounds are stable under approximations of the posterior in the total variation and $W_2$-Wasserstein metrics. Given sufficient computing resources, this yields bounds for LMC algorithms.

- We give high-probability data-dependent bounds on the test error, both for a hypothesis drawn from the Gibbs posterior and for the posterior mean, assuming that we can freely draw samples from the posterior. These bounds hold for the entire range of temperatures.

Our method is based on a combination of the PAC-Bayesian bounds (McAllester, 1999; Alquier, 2021; Rivasplata et al., 2020) with an integral representation of the log-partition function. This makes it possible to bound the logarithm of the density of the Gibbs posterior at a given temperature in terms of empirical averages at higher temperatures. A qualitative conclusion is that generalization in the under-regularized low-temperature regime ($\beta > n$) is already indicated by small training errors in the over-regularized high-temperature regime ($\beta < n$), where $n$ is the number of training examples.

### 1.1 RELATED LITERATURE

Many papers address the generalization of the Gibbs algorithm and Langevin Monte Carlo, with special focus on SGLD, which is the most popular algorithm. Most similar to this work is Raginsky et al. (2017), which bounds the distance to the Gibbs posterior and then its generalization error. Their bound, however, applies only to the high temperature regime $\beta < n$.

Several works concentrate on the optimization path of SGLD. Mou et al. (2018) gives both stability and PAC-Bayesian bounds. Pensia et al. (2018) applies the information theoretic generalization bounds of Xu & Raginsky (2017). These ideas are further developed by Negrea et al. (2019), where random subsets of the training data are used to define data-dependent priors. Farghly & Rebeschini (2021) gives time-independent bounds for SGLD, which are further improved by Futami & Fujisawa (2024). Most of the bounds in the above papers are in expectation. The very recent paper of Harel et al. (2025) gives a very elegant argument for Markov chain algorithms based on the second law of thermodynamics. If the invariant distribution is the Gibbs posterior, the bound along the entire optimization path is of order $\sqrt{\beta/n}$ but improvable to $\beta/n$.

Some papers give similar bounds for the Gibbs posterior, roughly of the form $\beta/n$ or $\sqrt{\beta/n}$ (Raginsky et al., 2017; Dziugaite & Roy, 2018; Kuzborskij et al., 2019; Rivasplata et al., 2020) or Maurer (2024) and Harel et al. (2025)). These bounds hold equally for random labels and are therefore vacuous for overparametrized hypothesis spaces in the low temperature regime $\beta > n$. To our knowledge, ours is the only bound for the Gibbs posterior, which is valid in this regime.

Other bounds have been developed for specific algorithms designed to optimize them. The milestone paper by Dziugaite & Roy (2017) is the most prominent example, and (Dziugaite & Roy, 2018) and (Pérez-Ortiz et al., 2021) are also in this category. Our bounds by contrast apply to the Gibbs posterior and LMC in their standard forms.

## 2 PRELIMINARIES

The relative entropy of two Bernoulli variables with expectations $p$ and $q$ is denoted

$$\kappa(p, q) = p \ln \frac{p}{q} + (1 - p) \ln \frac{1 - p}{1 - q}. \tag{1}$$

We also define the function $\kappa^{-1} : [0, 1] \times [0, \infty) \to [0, 1]$ by

$$\kappa^{-1}(p, t) = \inf \{q : q \geq p, \kappa(p, q) \geq t\}.$$

Throughout the following $(\mathcal{X}, \Sigma)$ is a measurable space of *data* with probability measure $\mu$. The i.i.d. random vector $\mathbf{x} \sim \mu^n$ is the training sample.

We let $(\mathcal{H}, \Omega)$ be a measurable space of *hypotheses*, and let $\ell : \mathcal{H} \times \mathcal{X} \to [0, \infty)$ be a prescribed loss function. Members of $\mathcal{H}$ are denoted $h$ or $g$. We write $L(h) := \mathbb{E}_{x \sim \mu}[\ell(h, x)]$ and $\hat{L}(h, \mathbf{x}) := (1/n) \sum_i \ell(h, x_i)$ respectively for the true (expected) and empirical error of hypothesis $h \in \mathcal{H}$. The set of probability measures on $(\mathcal{H}, \Omega)$ is denoted $\mathcal{P}(\mathcal{H})$.

A stochastic algorithm is a function $\nu : \mathcal{X}^n \to \mathcal{P}(\mathcal{H})$, which assigns to a training sample $\mathbf{x}$ a probability measure $\nu(\mathbf{x}) \in \mathcal{P}(\mathcal{H})$. The KL-divergence between two probability measures is the function KL $: (\rho, \nu) \in \mathcal{P}(\mathcal{H}) \times \mathcal{P}(\mathcal{H}) \mapsto \mathbb{E}_{h \sim \rho}\left[\ln \frac{d\rho}{d\nu}\right]$ if $\rho$ absolutely continuous w.r.t. $\nu$, otherwise the value is $\infty$. The total variation distance is defined as $d_{TV} : (\rho, \nu) \in \mathcal{P}(\mathcal{H}) \times \mathcal{P}(\mathcal{H}) \mapsto \sup_{A \in \Omega} |\rho(A) - \nu(A)|$. The $W_p$-Wasserstein distance is $W_p(\rho, \nu) = (\inf_W \mathbb{E}_{(x,y) \sim W}[\|x - y\|^p])^{1/p}$ with the infimum over all probability measures on $\mathcal{P}(\mathcal{H} \times \mathcal{H})$ with $\rho$ and $\nu$ as marginals.

There is an a-priori reference measure $\pi \in \mathcal{P}(\mathcal{H})$, called the *prior*. With a fixed prior, the Gibbs algorithm at inverse temperature $\beta > 0$ is the stochastic algorithm $G_\beta : \mathbf{x} \in \mathcal{X}^n \mapsto G_\beta(\mathbf{x}) \in \mathcal{P}(\mathcal{H})$

defined by

$$G_\beta\left(\mathbf{x}\right)\left(A\right) = \frac{1}{Z_\beta\left(\mathbf{x}\right)} \int_A e^{-\beta\hat{L}(h,\mathbf{x})} d\pi\left(h\right) \text{ for } A \in \Omega.$$

$G_\beta\left(\mathbf{x}\right)$ is called the *Gibbs-posterior,* the normalizing factor

$$Z_\beta\left(\mathbf{x}\right) := \int_\mathcal{H} e^{-\beta\hat{L}(h,\mathbf{x})} d\pi\left(h\right)$$

is called the *partition function.* The motivation for the Gibbs posterior is that it puts larger weights on hypotheses with smaller empirical error.

Given a stochastic algorithm $\nu$ we define a probability measure $\rho_\nu$ on $\mathcal{H} \times \mathcal{X}^n$ by

$$\rho_\nu\left(A\right) = \mathbb{E}_{\mathbf{x}\sim\mu^n}\mathbb{E}_{h\sim\nu(\mathbf{x})}\left[1_A\left(h,\mathbf{x}\right)\right] \text{ for } A \in \Omega \otimes \Sigma^{\otimes n}. \tag{2}$$

Then, $\mathbb{E}_{(h,\mathbf{x})\sim\rho_\nu}\left[\phi\left(h,\mathbf{x}\right)\right] = \mathbb{E}_\mathbf{x}\mathbb{E}_{h\sim\nu(\mathbf{x})}\left[\phi\left(h,\mathbf{x}\right)\right]$ for measurable $\phi : \mathcal{H} \times \mathcal{X}^n \to \mathbb{R}$. To draw the pair $(h,\mathbf{x})$ from $\rho_\nu$ we first draw the training sample $\mathbf{x}$, and then sample $h$ from $\nu\left(\mathbf{x}\right)$. The main objective in learning is that the risk $\mathbb{E}_{x\sim\mu}\left[f\left(h,x\right)\right]$ is small with high probability in $(h,\mathbf{x}) \sim \rho_\nu$, where $f$ is some application-dependent loss function, possibly different from $\ell$. In the sequel we will give corresponding guarantees.

## 3 BOUNDS FOR THE GIBBS POSTERIOR

In this section, we make the idealized assumption that we are free to sample from the Gibbs posterior at any finite $\beta \geq 0$.

### 3.1 AN INTEGRAL REPRESENTATION OF THE FREE ENERGY

**Lemma 3.1.** *Let* $0 = \beta_0 < \beta_1 < \cdots < \beta_K = \beta$*. Then*

$$-\ln Z_\beta(\mathbf{x}) = \int_0^\beta \mathbb{E}_{h\sim G_\gamma(\mathbf{x})}\left[\hat{L}(h,\mathbf{x})\right]d\gamma \leq \sum_{k=1}^K (\beta_k - \beta_{k-1})\mathbb{E}_{g\sim G_{\beta_{k-1}}(\mathbf{x})}\left[\hat{L}(g,\mathbf{x})\right].$$

*Proof.* Let $A(\beta) = -\ln Z_\beta(\mathbf{x})$. One verifies the identities

$$\begin{aligned}
A(0) &= 0, \\
A'(\beta) &= \frac{1}{Z_{\beta,\pi}(\mathbf{x})} \int_\mathcal{H} \hat{L}(h,\mathbf{x})e^{-\beta\hat{L}(h,\mathbf{x})}d\pi(h) = \mathbb{E}_{h\sim G_\beta(\mathbf{x}}\left[\hat{L}(h,\mathbf{x})\right], \\
A''(\beta) &= -\left(\mathbb{E}_{h\sim G_\beta(\mathbf{x})}\left[\hat{L}(h,\mathbf{x})^2\right] - \left(\mathbb{E}_{h\sim G_\beta(\mathbf{x})}\left[\hat{L}(h,\mathbf{x})\right]\right)^2\right) \leq 0.
\end{aligned}$$

The equality in the lemma then follows from the first two identities above and the fundamental theorem of calculus, and the inequality follows from the last identity, which shows that $\mathbb{E}_{g\sim G_{\beta_{k-1},\pi}(\mathbf{x})}\left[\hat{L}(g,\mathbf{x})\right]$ is non-increasing in $\beta$. $\qquad\square$

In statistical physics there is a formal analogy, where the function $h \mapsto \hat{L}(h,\mathbf{x})$ is the ($\mathbf{x}$-dependent) energy of the system in the state $h$, and $\beta$ is the inverse temperature. The Gibbs posterior then becomes the "canonical ensemble" (Gibbs, 1902), describing the probability of states in equilibrium with a heat bath at temperature $\beta^{-1}$. The function $\beta \mapsto A(\beta)$ plays an important role: $\beta^{-1}A(\beta)$ is the Helmholz free energy, $A'(\beta) = \mathbb{E}_{h\sim G_\beta(\mathbf{x})}\left[\hat{L}(h,\mathbf{x})\right]$ is the thermal average of the energy, $-\beta A'(\beta) + A(\beta)$ is the entropy and $-A''(\beta)$ is proportional to the heat capacity at temperature $\beta^{-1}$ (see e.g. Huang, 2008).

For $h \in \mathcal{H}$, $\mathbf{x} \in \mathcal{X}^n$ and an increasing sequence $\boldsymbol{\beta} = (\beta_1 < \cdots < \beta_K)$ of positive numbers, we denote

$$\Gamma(h,\mathbf{x},\boldsymbol{\beta}) = -\beta_K\hat{L}(h,\mathbf{x}) + \sum_{k=1}^K (\beta_k - \beta_{k-1})\mathbb{E}_{g\sim G_{\beta_{k-1}}(\mathbf{x})}\left[\hat{L}(g,\mathbf{x})\right]. \tag{3}$$

So Lemma 3.1 states that, when $\beta_0 = 0$, then,

$$-\beta_K\hat{L}\left(h,\mathbf{x}\right) - \ln Z_{\beta_K}\left(\mathbf{x}\right) \leq \Gamma\left(h,\mathbf{x},\boldsymbol{\beta}\right). \tag{4}$$

Note that $\Gamma\left(h,\mathbf{x},\boldsymbol{\beta}\right)$ depends *only* on the training data $\mathbf{x}$, the sequence $\boldsymbol{\beta}$ and the hypothesis $h$.

### 3.2 BOUNDS

The function $F$ in the following is a placeholder for a random variable related to the generalization gap, which we would like to bound with high probability.

**Theorem 3.2.** *Let $F : \mathcal{H} \times \mathcal{X}^n \to \mathbb{R}$ be some measurable function, $\beta > 0$ and $\boldsymbol{\beta} = (\beta_1 < \cdots < \beta_K)$ as above with $\beta_0 = 0$ and $\beta_K = \beta$. Then,*

*(i) for $\delta > 0$ with probability at least $1 - \delta$ in $\mathbf{x} \sim \mu^n$ and $h \sim G_\beta(\mathbf{x})$*

$$F(h, \mathbf{x}) \leq \Gamma(h, \mathbf{x}, \boldsymbol{\beta}) + \ln \mathbb{E}_{\mathbf{x}} \mathbb{E}_{g \sim \pi} \left[ e^{F(g, \mathbf{x})} \right] + \ln(1/\delta),$$

*(ii) for $\delta > 0$ with probability at least $1 - \delta$ in $\mathbf{x} \sim \mu^n$*

$$\mathbb{E}_{h \sim G_\beta(\mathbf{x})} \left[ F(h, \mathbf{x}) \right] \leq \mathbb{E}_{h \sim G_\beta(\mathbf{x})} \left[ \Gamma(h, \mathbf{x}, \boldsymbol{\beta}) \right] + \ln \mathbb{E}_{\mathbf{x}} \mathbb{E}_{g \sim \pi} \left[ e^{F(g, \mathbf{x})} \right] + \ln(1/\delta).$$

*Proof.* By Markov's inequality, for any real random variable $Y$

$$\Pr\left\{ Y > \ln \mathbb{E}\left[ e^Y \right] + \ln(1/\delta) \right\} = \Pr\left\{ e^Y > \mathbb{E}\left[ e^Y \right]/\delta \right\} \leq \delta.$$

To prove (i), we apply this to the random variable $Y = F(h, \mathbf{x}) + \beta \hat{L}(h, \mathbf{x}) + \ln Z_\beta(\mathbf{x})$ on the probability space $\left( \mathcal{H} \times \mathcal{X}^n, \Omega \otimes \Sigma^{\otimes n}, \rho_{G_\beta} \right)$ as defined in (2). Together with the definition of the Gibbs posterior, this gives, with probability at least $1 - \delta$ in $(h, \mathbf{x}) \sim \rho_{G_\beta}$ (equivalent to saying $\mathbf{x} \sim \mu^n$ and $h \sim G_\beta(\mathbf{x})$),

$$F(h, \mathbf{x}) + \beta \hat{L}(h, \mathbf{x}) + \ln Z_\beta(\mathbf{x})$$

$$\leq \ln \mathbb{E}_{\mathbf{x}} \mathbb{E}_{g \sim G_\beta(\mathbf{x})} \left[ e^{F(g, \mathbf{x}) + \beta \hat{L}(g, \mathbf{x}) + \ln Z_\beta(\mathbf{x})} \right] + \ln(1/\delta)$$

$$= \ln \mathbb{E}_{\mathbf{x}} \mathbb{E}_{g \sim \pi} \left[ e^{F(g, \mathbf{x}) + \beta \hat{L}(g, \mathbf{x}) + \ln Z_\beta(\mathbf{x}) - \beta \hat{L}(g, \mathbf{x}) - \ln Z_\beta(\mathbf{x})} \right] + \ln(1/\delta)$$

$$= \ln \mathbb{E}_{\mathbf{x}} \mathbb{E}_{g \sim \pi} \left[ e^{F(g, \mathbf{x})} \right] + \ln(1/\delta).$$

Subtract $\beta \hat{L}(h, \mathbf{x}) + \ln Z_\beta(\mathbf{x})$ and use (4). For (ii) apply Markov's inequality to $\mathbb{E}_{h \sim G_\beta(\mathbf{x})} \left[ F(h, \mathbf{x}) + \beta \hat{L}(h, \mathbf{x}) + \ln Z_\beta(\mathbf{x}) \right]$ instead. By Jensen's inequality

$$e^{\mathbb{E}_{h \sim G_\beta(\mathbf{x})} \left[ F(h, \mathbf{x}) + \beta \hat{L}(h, \mathbf{x}) + \ln Z_\beta(\mathbf{x}) \right]} \leq \mathbb{E}_{g \sim G_\beta(\mathbf{x})} \left[ e^{F(g, \mathbf{x}) + \beta \hat{L}(g, \mathbf{x}) + \ln Z_\beta(\mathbf{x})} \right]$$

and proceed as above using (4). $\qquad \square$

Up to the application of (4), the above proof of (i) just gives the single-draw version of the PAC-Bayesian bound as in Rivasplata et al. (2020) applied to the Gibbs posterior, while (ii) is the standard PAC-Bayesian bound applied to the Gibbs posterior until (4) is invoked.

### 3.3 LOSS FUNCTIONS AND SECONDARY LOSS FUNCTIONS

To apply Theorem 3.2, we need to control the exponential moment $\mathbb{E}_{\mathbf{x}} \mathbb{E}_{g \sim \pi} \left[ e^{F(g, \mathbf{x})} \right]$, but otherwise we have free choice of the function $F$. This gives the method some flexibility. By Tonelli's Theorem we can exchange the two expectations, and often there is a bound on $\mathbb{E}_{\mathbf{x}} \left[ e^{F(g, \mathbf{x})} \right]$ uniform in $g$, which then carries over to $\mathbb{E}_{g \sim \pi} \mathbb{E}_{\mathbf{x}} \left[ e^{F(g, \mathbf{x})} \right]$, since $\pi$ is a probability measure. In this way bounds for sub-Gaussian or sub-exponential losses can be obtained, but also for U-statistics or even non-i.i.d. data, sampled from the trajectories of time-homogeneous, ergodic Markov chains. In Section B.1 in the appendix, we derive a bound for sub-Gaussian losses from Theorem 3.2; other examples are planned for a longer version of the paper.

The function $F$ may be defined in terms of other, application-dependent loss functions, which are different from the loss $\ell$, which defines the Gibbs posterior and the functional $\Gamma$. To illustrate this point, let $f : \mathcal{H} \times \mathcal{X} \to [0, 1]$ be measurable and set $F(h, \mathbf{x}) = n \, \kappa \left( \frac{1}{n} \sum_i f(h, x_i), \mathbb{E}_x \left[ f(h, x) \right] \right)$, with $\kappa$ the relative entropy as in (1). Then, Theorem 1 of Maurer (2004) gives $\mathbb{E}_{\mathbf{x}} \left[ e^{F(h, \mathbf{x})} \right] \leq 2\sqrt{n}$ for $n \geq 8$. Substitution in Theorem 3.2 and division by $n$ then give the following corollary.

**Corollary 3.3.** *Let $f : \mathcal{H} \times \mathcal{X} \to [0,1]$ be measurable, $\delta > 0$ and $n \geq 8$. Then, with probability at least $1 - \delta$ in $\mathbf{x} \sim \mu^n$ and $h \sim G_\beta(\mathbf{x})$*

$$\kappa \left( \frac{1}{n} \sum_i f(h, x_i), \mathbb{E}_x[f(h, x)] \right) \leq \frac{1}{n} \left( \Gamma(h, \mathbf{x}, \boldsymbol{\beta}) + \ln \left( \frac{2\sqrt{n}}{\delta} \right) \right)$$

*and with probability at least $1 - \delta$ in $\mathbf{x} \sim \mu^n$*

$$\kappa \left( \frac{1}{n} \sum_i \mathbb{E}_{h \sim G_\beta(\mathbf{x})} \left[ f(h, x_i) \right], \mathbb{E}_{h \sim G_\beta(\mathbf{x})} \mathbb{E}_x \left[ f(h, x) \right] \right) \leq \frac{1}{n} \left( \Gamma(h, \mathbf{x}, \boldsymbol{\beta}) + \ln \left( \frac{2\sqrt{n}}{\delta} \right) \right).$$

For the second part, we used the joint convexity of $\kappa$. Under the conditions of this corollary, the second inequality becomes

$$\mathbb{E}_{h \sim G_\beta(\mathbf{x})} \mathbb{E}_x \left[ f(h, x) \right] \leq \kappa^{-1} \left( \frac{1}{n} \sum_i \mathbb{E}_{h \sim G_\beta(\mathbf{x})} \left[ f(h, x_i) \right], \frac{1}{n} \left( \Gamma(h, \mathbf{x}, \beta) + \ln \left( \frac{2\sqrt{n}}{\delta} \right) \right) \right) \quad (5)$$

with an analogous version for the single-draw case. This is how we compute bounds in our experiments. For an illustration, please refer to C.2.1.

Here $f$ plays the role of a secondary loss function, typically different from the loss function $\ell$, which defines the Gibbs posterior. In applications one would define the Gibbs posterior in terms of a differentiable, potentially unbounded loss function $\ell$ and approximate it with a suitable Monte Carlo method. For classification, however, one is interested in bounding the 0-1 loss obtained by some threshold on $\ell$. For binary classification each $x \in \mathcal{X}$ is of the form $x = (z, y)$, where $y \in \{-1, 1\}$ is the label corresponding to features $z$ and $f(h, (z, y)) := 1_{(-\infty, 0)}(y \ell(h, z))$ is the 0-1 loss, which can be directly substituted in (5) above to bound the misclassification probability in terms of its empirical counterpart.

## 4  BOUNDS FOR LANGEVIN MONTE CARLO

For this section, we assume $\mathcal{H} = \mathbb{R}^d$ and an isotropic Gaussian prior $\pi$ of width $\sigma$. We condition on the training data $\mathbf{x}$, reference to which we omit. The Gibbs posterior is an idealization, from which it is impossible to sample directly. Nevertheless a number of works (Raginsky et al., 2017; Dalalyan & Karagulyan, 2017; Brosse et al., 2018; Vempala & Wibisono, 2019; Dwivedi et al., 2019; Nemeth & Fearnhead, 2021; Balasubramanian et al., 2022) discuss algorithms (SGLD, ULA, MALA, etc, here summarized as Langevin Monte Carlo (LMC)), capable of approximating a probability measure $\nu$ on $\mathbb{R}^d$ of the form $\nu \propto \exp(-V)$ or some nearby limiting distribution. In the following, we discuss one of these algorithms.

### 4.1  ULA

We focus on the results of Vempala & Wibisono (2019), which do not require convexity of $V$ and instead assume that the measure $\nu$ satisfies a log-Sobolev inequality (LSI) in the sense that for all smooth $f : \mathbb{R}^d \to \mathbb{R}$

$$\mathbb{E}_{h \sim \nu} \left[ f^2(h) \ln f^2(h) \right] - \mathbb{E}_{h \sim \nu} \left[ f^2(h) \right] \ln \mathbb{E}_{h \sim \nu} \left[ f^2(h) \right] \leq \frac{2}{\alpha} \mathbb{E}_{h \sim \nu} \left[ \| (\nabla f)(h) \|^2 \right] \quad (6)$$

for some $\alpha > 0$. An LSI is satisfied when $V$ is strongly convex, but, importantly, also for measures which are bounded perturbations of measures satisfying an LSI (Holley & Stroock (1986)). Vempala & Wibisono (2019) give further examples and a list of references for measures, which are not log-concave and satisfy an LSI. Raginsky et al. (2017) show, that under dissipativity conditions of the loss the Gibbs posterior $G_\beta(\mathbf{x})$ satisfies an LSI with constant independent of $\mathbf{x}$.

Consider the iterative algorithm

$$h_{t+1} = h_t - \epsilon \nabla V(h_t) + \sqrt{2\epsilon} \xi_t, \quad (7)$$

where $\epsilon$ is a step size, the $\xi_t \sim \mathcal{N}(0, I)$ are independent Gaussian vectors and $h_0$ is drawn from some initial distribution $\nu_0$. Some authors call this algorithm simply LMC, for Langevin Monte Carlo. We

call it ULA, alongside Durmus & Moulines (2017),Dwivedi et al. (2019) and Vempala & Wibisono (2019), for Un-adjusted Langevin Algorithm, because it misses the Metropolis-type accept-reject step, which would guarantee that the invariant distribution is indeed the Gibbs posterior. A popular variant of ULA is Stochastic Gradient Langevin Dynamics (SGLD) (Welling & Teh, 2011; Raginsky et al., 2017) where the gradient is replaced by an unbiased estimate, typically realized with random minibatches. Here, we restrict ourselves to ULA with a constant step size, because it has the least number of parameters to adjust, but in experiments we also use the computationally more efficient SGLD.

As $\epsilon \to 0$, ULA recovers the Continuous Langevin Dynamics (CLD) given by the stochastic differential equation

$$dh_t = -\nabla V(h_t)\, dt + \sqrt{2}dB_t,$$

where $B_t$ is centered standard Brownian motion in $\mathbb{R}^d$. CLD converges exponentially to the Gibbs posterior (Chiang et al., 1987). For $\epsilon > 0$, the distribution $\nu_{\epsilon,t}$ of ULA converges as $t \to \infty$ to a biased limiting distribution $\nu_\epsilon$ which is generally different from $\nu$, but expected to be closer to $\nu$ as $\epsilon$ becomes smaller. Vempala & Wibisono (2019) use the LSI assumption and coupling to control the difference between CLD and ULA along their path and prove the following result.

**Theorem 4.1.** *Assume that $\nu$ satisfies the log-Sobolev inequality (6) with $\alpha > 0$, that the Hessian of $V$ satisfies $-LI \preceq \nabla^2 V(h) \preceq LI$ for all $h$ and some $L < \infty$, and that $0 < \epsilon \leq \alpha/\left(4L^2\right)$. Then, for $t \geq 0$*

$$KL\left(\nu, \nu_{\epsilon,t}\right) \leq e^{-\alpha\epsilon t}KL\left(\nu, \nu_0\right) + \frac{8\epsilon dL^2}{\alpha}.$$

The first exponential term is due to the mismatch of the initial distribution and $\nu$. The second term bounds the divergence between the limiting distribution $\nu_\epsilon$ and $\nu$. Similar results exist under different conditions on the potential $V$. Cheng et al. (2018) for example require $V$ to be strongly convex outside of a ball instead of the log-Sobolev inequality and gives bounds in terms of the $W_1$-Wasserstein metric. Raginsky et al. (2017) give bounds for $W_2$ under dissipativity assumptions. The next corollary adapts Theorem 4.1 to the situation studied in this paper.

**Corollary 4.2.** *For $\beta > 0$ consider the Gibbs posterior $G_\beta$ corresponding to $\hat{L}(h)$, with centered Gaussian prior of width $\sigma$. Assume that it satisfies the log-Sobolev inequality (6) with $\alpha > 0$, that the Hessian of $\hat{L}$ satisfies $-RI \preceq \nabla^2\hat{L}(h) \preceq RI$ for all $h$ and some $R < \infty$, and that $0 < \eta \leq \alpha/\left(4\left(\beta R + \frac{1}{\sigma^2}\right)^2\right)$. Consider the algorithm*

$$h_{t+1} = h_t - \eta\nabla_h\hat{L}(h_t) - \frac{\eta h_t}{\beta\sigma^2} + \sqrt{\frac{2\eta}{\beta}}\xi_t, \tag{8}$$

*where $h_0 \sim \nu_0$ and the $\xi_t \sim \mathcal{N}(0, I)$ are independent Gaussian random variables. Let $D(\beta) = KL\left(G_\beta, \nu_0\right)$ and let $\nu_{\beta,\eta,t}$ be the distribution of $h_t$ after $t$ steps. Then,*

*(i) $KL\left(G_\beta, \nu_{\beta,\eta,t}\right) \leq e^{-\alpha\eta t/\beta}D(\beta) + \frac{8\eta d}{\beta\alpha}\left(\beta R + \frac{1}{\sigma^2}\right)^2.$*

*(ii) $W_2\left(G_\beta, \nu_{\beta,\eta,t}\right) \leq \frac{2}{\alpha}e^{-\alpha\eta t/\beta}D(\beta) + \frac{16\eta d}{\beta\alpha^2}\left(\beta R + \frac{1}{\sigma^2}\right)^2.$*

*(iii) $d_{TV}\left(G_\beta, \nu_{\beta,\eta,t}\right) \leq e^{-\alpha\eta t/(2\beta)}\sqrt{D(\beta)} + 2\sqrt{\frac{\eta d}{\beta\alpha}}\left(\beta R + \frac{1}{\sigma^2}\right).$*

*Proof.* (i) follows directly from Theorem 4.1 and the substitutions $V(h) = \beta\hat{L}(h) + \|h\|^2/\left(2\sigma^2\right)$, $\epsilon = \eta/\beta$ and $L = \beta R + \frac{1}{\sigma^2}$. Then $\nu = G_\beta$ with Gaussian prior of width $\sigma$, and ULA becomes (8). (ii) follows from Theorem 1 of Otto & Villani (2000) and the LSI assumption, and (iii) follows from Pinsker's inequality (see e.g. Boucheron et al. (2013), Theorem 4.19). $\qquad\square$

### 4.2 STABILITY OF THE BOUNDS

We now show the stability of our bounds for approximation in total variation and $W_2$-Wasserstein metrics, under boundedness or Lipschitz conditions. Together with Corollary 4.2 this implies bounds for the algorithm defined in (8).

We assume that there is a target approximation $\nu_\beta(\mathbf{x})$ of $G_\beta(\mathbf{x})$, for which we want to compute a high probability bound, either for the single draw version on $F(h, \mathbf{x})$ as $\mathbf{x} \sim \mu^n$ and $h \sim \nu_\beta(\mathbf{x})$, or, for the classical PAC-Bayesian version, on $\mathbb{E}_{h \sim \nu_\beta(\mathbf{x})}[F(h, \mathbf{x})]$ as $\mathbf{x} \sim \mu^n$. It is not surprising that the single-draw bound will require a much closer approximation of the Gibbs posterior.

Since the bounding functional $\Gamma(h, \mathbf{x}, \boldsymbol{\beta})$ depends on the Gibbs posteriors $G_{\beta_k}(\mathbf{x})$ for $k \in \{1, \ldots, K-1\}$, we require corresponding approximations $\nu_{\beta_k}$ of $G_{\beta_k}(\mathbf{x})$ to compute the bound. To streamline notation, we define

$$\Gamma_\nu(h, \mathbf{x}, \boldsymbol{\beta}) = -\beta \hat{L}(h, \mathbf{x}) + \sum_{k=1}^{K} (\beta_k - \beta_{k-1}) \mathbb{E}_{g \sim \nu_{\beta_{k-1}}(\mathbf{x})} [\hat{L}(g, \mathbf{x})]$$

for $0 = \beta_0 < \beta_1 < \cdots < \beta_K = \beta$ and $(\nu_0(\mathbf{x}), \nu_{\beta_1}(\mathbf{x}), \cdots, \nu_{\beta_{k-1}}(\mathbf{x})) \in \mathcal{P}(\mathcal{H})^K$. Like $\Gamma(h, \mathbf{x}, \beta)$, the functional $\Gamma_\nu(h, \mathbf{x}, \beta)$ depends on the training data, but it can also be computed by repeated execution of the algorithm (8). The next theorem states the obtained bound in terms of the approximation errors in total variation.

**Theorem 4.3.** *Suppose that $\mathcal{H} = \mathbb{R}^d$ and that there are numbers $m, M < \infty$ such that for every $\mathbf{x}$ in $\mathcal{X}^n$ and $h \in \mathcal{H}$ we have $|\ell(h, \mathbf{x})| \leq m$ and $|F(h, \mathbf{x})| \leq M$. Let $0 = \beta_0 < \beta_1 < \cdots < \beta_K = \beta$ and $\nu_{\beta_k}(\mathbf{x}) \in \mathcal{P}(\mathcal{H})$ be such that $d_{TV}(\nu_{\beta_k}(\mathbf{x}), G_{\beta_k}(\mathbf{x})) = \epsilon_{\beta_k}$. Then,*

*(i) with probability at least $1 - \delta$ as $\mathbf{x} \sim \mu^n$ and $h \sim \nu_\beta(\mathbf{x})$*

$$F(h, \mathbf{x}) \leq \Gamma_\nu(h, \mathbf{x}, \boldsymbol{\beta}) + \ln \mathbb{E}_\mathbf{x} \mathbb{E}_{h \sim \pi} \left[ e^{F(h, \mathbf{x})} \right] + \ln \frac{1}{\delta} + \ln \left( 2 e^{M + \beta m} \epsilon_\beta \right) + \sum_{k=1}^{K} (\beta_k - \beta_{k-1}) m \epsilon_{\beta_{k-1}}.$$

*(ii) with probability at least $1 - \delta$ as $\mathbf{x} \sim \mu^n$*

$$\mathbb{E}_{h \sim \nu_\beta(\mathbf{x})}[F(h, \mathbf{x})] \leq \mathbb{E}_{h \sim \nu_\beta(\mathbf{x})}[\Gamma_\nu(h, \mathbf{x}, \boldsymbol{\beta})] + \ln \mathbb{E}_\mathbf{x} \mathbb{E}_{h \sim \pi} \left[ e^{F(h, \mathbf{x})} \right] + \ln \frac{1}{\delta}$$

$$+ (M + \beta m) \epsilon_\beta + \sum_{k=1}^{K} (\beta_k - \beta_{k-1}) m \epsilon_{\beta_{k-1}}.$$

The proof, given in Section B.2, is similar to that of Theorem 3.2 and applies Markov's inequality with $\nu_\beta$ instead of $G_\beta$. It then uses the fact that, if $f$ is a bounded measurable function, then $|\mathbb{E}_{\nu_1}[f] - \mathbb{E}_{\nu_2}[f]| \leq \|f\|_\infty d_{TV}(\nu_1, \nu_2)$. For the single-draw version (i) this is applied to $\left( \mathbb{E}_{\nu_\beta} - \mathbb{E}_{G_\beta} \right) \left[ e^{F(h, \mathbf{x}) + \beta \hat{L}(h, \mathbf{x}) + \ln Z_\beta(\mathbf{x})} \right]$, which causes the exponential dependence on $\beta m$ and $M$. For (ii) we can apply this in the exponent to $\left( \mathbb{E}_{\nu_\beta} - \mathbb{E}_{G_\beta} \right) \left[ F(h, \mathbf{x}) + \beta \hat{L}(h, \mathbf{x}) \right]$, and the logarithm makes the dependence linear. The rest of the proof is mechanical.

The last terms in (i) and (ii) are the additional errors due to the approximations of the Gibbs posteriors. The worst term is clearly the first one in (i) due to the exponential dependence on the proxy function $F$, which is typically of order $n$ and on $\beta$, which is larger than $n$ in the regime in which we are interested. In practice, the requirement of such an approximation is prohibitive for the single-draw version. The bound in (ii) has more moderate approximation requirements. In this case, we can also give bounds in terms of the $W_2$-Wasserstein metric (as guaranteed by Corollary 4.2), if $F$ and $\ell$ satisfy a Lipschitz condition instead of boundedness. We will use the following fact: Since $W_1 \leq W_2$ it follows from the Kantorovich-Rubinstein Theorem (Villani, 2009), that for any real Lipschitz function $f$ on $\mathcal{H}$ and probability measures $\nu_1, \nu_2 \in \mathcal{P}(\mathcal{H})$

$$|\mathbb{E}_{h \sim \nu_1}[f(h)] - \mathbb{E}_{h \sim \nu_2}[f(h)]| \leq \|f\|_{\text{Lip}} W_1(\nu_1, \nu_2) \leq \|f\|_{\text{Lip}} W_2(\nu_1, \nu_2),$$

where $\|.\|_{\text{Lip}}$ is the Lipschitz-seminorm. The following result is then immediate, with proof exactly as in (ii) of Theorem 4.3.

**Theorem 4.4.** *Assume the conditions of Theorem 4.3, except that instead of $|\ell(h, \mathbf{x})| \leq m$ and $|F(h, \mathbf{x})| \leq M$ we have $\|\ell(., \mathbf{x})\|_{\text{Lip}} \leq m$ and $\|F(., \mathbf{x})\|_{\text{Lip}} \leq M$ and that $W_2(\nu_{\beta_k}(\mathbf{x}), G_\beta(\mathbf{x})) = \epsilon_{\beta_k}$. Then, with probability at least $1 - \delta$ as $\mathbf{x} \sim \mu^n$*

$$\mathbb{E}_{h \sim \nu_\beta(\mathbf{x})}[F(h, \mathbf{x})] \leq \mathbb{E}_{h \sim \nu_\beta(\mathbf{x})}[\Gamma_\nu(h, \mathbf{x}, \boldsymbol{\beta})] + \ln \mathbb{E}_\mathbf{x} \mathbb{E}_{h \sim \pi} \left[ e^{F(h, \mathbf{x})} \right] + \ln \frac{1}{\delta}$$

$$+ (M + \beta m) \epsilon_\beta + \sum_{k=1}^{K} (\beta_k - \beta_{k-1}) m \epsilon_{\beta_{k-1}}.$$

## 5 EXPERIMENTS

The purpose of our experiments is to show that our method gives nontrivial bounds on the test error for real-world data, while correctly bounding the test error on impossible data, where the same algorithm also achieves a small training error. The real-world data are either the MNIST dataset, subdivided into the two classes of characters 0-4 and 5-9, or the CIFAR-10 dataset to distinguish between animals and vehicles. For impossible data, we randomize the labels of the training data. Our experiments are computationally heavy, so we generally use small sample sizes, from 2000 to 8000 examples. The hypothesis space is the set of weight vectors for a neural network with ReLU activation functions constrained by a Gaussian prior distribution with $\sigma = 5$. Neural network architectures are described in Section C.1.1 of the appendix. To approximately sample the weight vectors in the vicinity of the Gibbs posterior, we use ULA as in (8) or SGLD (Welling & Teh, 2011) with constant step size $\eta$. To ensure reproducibility, we provide the code and experimental results in an anonymous repository at `https://anonymous.4open.science/r/Gibbs-Generalization-45F1`.

### 5.1 THE LOSS FUNCTION $\ell$

Most experiments were done with bounded loss functions $\ell$, either bounded binary cross-entropy as described in Appendix D of Dziugaite & Roy (2018) or the Savage loss (Masnadi-Shirazi & Vasconcelos, 2008). As unbounded loss function we tried binary cross-entropy (BCE) (Section C.2.7), but with a smaller value of $\sigma$, so as to avoid excessive training errors for small values of $\beta$. We compute bounds for the 0-1 loss, using the method described in Section 3.3.

### 5.2 APPROXIMATING THE ERGODIC MEAN

As we know of no sufficient criterion for convergence, we terminate iterations at time $T$, when a very slow running mean $\mathbb{M}_{\text{stop}}$ of the loss trajectory $\left(\hat{L}(h_{\beta_k t}, \mathbf{x})\right)_{t=0}^{T}$ stops decreasing. A second running mean $\mathbb{M}_{\text{erg}}$ is used as an approximation of the ergodic mean and thus of expectations in the invariant distribution. We thus replace all expectations $\mathbb{E}_{h \sim G_{\beta_k}}\left[\hat{L}(h, \mathbf{x})\right]$ occurring in the bounds by $\mathbb{M}_{\text{erg}}\left[\left(\hat{L}(h_{\beta_k t}, \mathbf{x})\right)_{t=0}^{T}\right]$. Both running means $\mathbb{M}_{\text{stop}}$ and $\mathbb{M}_{\text{erg}}$ are implemented as first-order, recursive lowpass filters described in Section C.1.3 of the appendix.

### 5.3 CALIBRATION

The theoretical bounds of Corollary 4.2 in combination with Theorem 4.3 can only serve as guidance for the computation of practical bounds, because the quantities $R$ and $\alpha$ are impossible to estimate in practice. But even if we assume these to be in the order of unity, the bounds are too coarse to distinguish between different temperatures with realistic stepsizes.

A simple calculation shows that $\text{KL}\left(G_\beta, G_{2\beta}\right) \leq \beta \left(\mathbb{E}_{G_\beta}\left[\hat{L}\right] - \mathbb{E}_{G_{2\beta}}\left[\hat{L}\right]\right) \leq \beta \mathbb{E}_{G_\beta}\left[\hat{L}\right]$ (Lemma B.5 in Appendix B.3). By Corollary 4.2 we should therefore have at least $8\eta d R^2/\alpha < \mathbb{E}_{G_\beta(\mathbf{x})}\left[\hat{L}\right]$ to distinguish between the expectations in the Gibbs posterior for $\beta$ and $2\beta$. The smallest neural network we use has $d = 392,500$. If $\ell$ has values in $[0, 1]$ then $\mathbb{E}_{G_\beta(\mathbf{x})}\left[\hat{L}\right] \leq 1$, so even if $R$ and $\alpha$ are set to 1, we would need stepsizes in the order of $10^{-7}$. Safe values of $\eta$, as suggested by the theoretical results in Section 4, are therefore impossible in practice, and the bound has to be adapted to a realistic choice of $\eta$.

The first possibility is to simply ignore the un-estimable error terms in Theorem 4.3, and to compute the bounds from $\Gamma_\nu\left(h, \mathbf{x}, \beta\right)$ in place of $\Gamma\left(h, \mathbf{x}, \beta\right)$ as in Theorem 3.2. We show some of these uncalibrated bounds, which already make nontrivial predictions, in Section C.2.6.

A second method uses a single calibration parameter, whose value is computed from the training data. We assume that the computed functional $\Gamma_\nu\left(h, \mathbf{x}, \beta\right)$ fails to estimate $\Gamma\left(h, \mathbf{x}, \beta\right)$ by a factor $r\left(\mathbf{x}\right) > 0$, which we define as the smallest factor of $\Gamma_\nu$, for which we obtain a correct upper bound on the 01-error with random labels for all the $\beta_k$. For a precise definition see Section B.4.

It is a purely experimental finding, that our choice of $r$ leads to correct and surprisingly tight upper bounds on the test error of correctly labeled data in all cases we tried. We emphasize that our calibration procedure depends only on the training data.

## 5.4 RESULTS

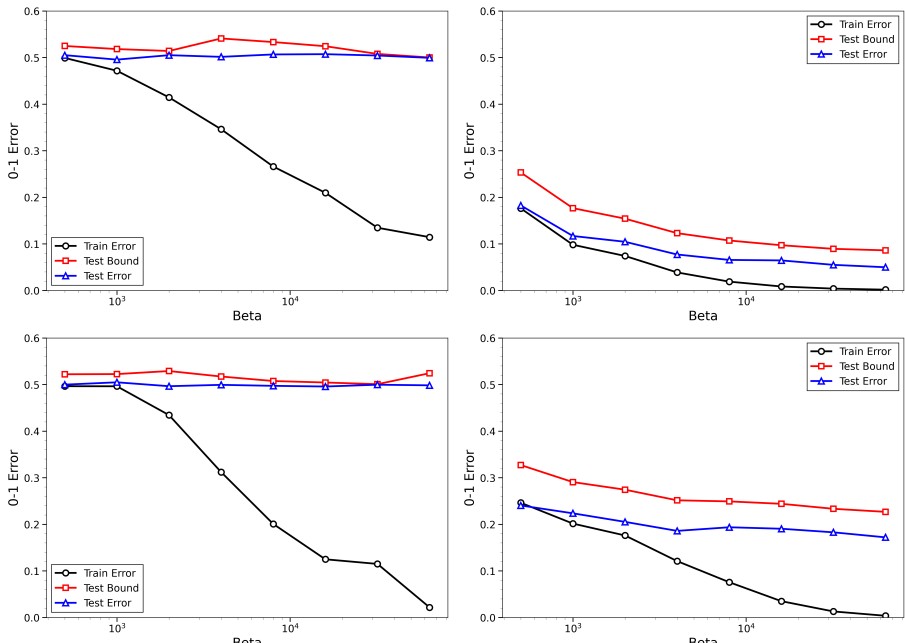

Figure 1: SGLD on MNIST and CIFAR-10 with 8000 training examples, MNIST above and CIFAR-10 below, random labels on the left, correct labels on the right. Both random and true labels are trained with the same algorithm and parameters on a fully connected ReLU network with two hidden layers of 1000 and 1500 units, respectively. The calibration factor for MNIST is 0.77, for CIFAR-10 0.89. Train error, test error and our bound for the Gibbs posterior average of the 0-1 loss are plotted against $\beta$.

Several experiments confirm the validity of the proposed bounds. An example is shown in Figure 1, where a fully connected ReLU-network with two hidden layers of 1000 (respectively 1500) units each is trained with SGLD at inverse temperatures $\beta = 0, 500, 1000, 2000, 4000, 8000, 16000, 32000$, and $64000$. The train error for random labels is about 0.1 (or even less) at $\beta = 64000$, where the bound is above 0.5. The test error for correct labels, however, is tightly bounded above.

Notice that for MNIST, which has the tightest bounds, the training error for the true labels is rapidly decreasing from 0.5 to 0.17 at $\beta = 500$ and to 0.1 at $\beta = 1000$. The more moderate initial decrease for CIFAR-10 corresponds to the tendency to overfit on this more difficult dataset. This confirms the intuition, that good generalization at low temperatures is already announced in the high temperature regime.

We generally found the uncalibrated bounds for ULA tighter than those for SGLD, consistent with the findings of Brosse et al. (2018). Experimental bounds for single draws from the posterior and various other experiments are reported in Section C.2.

## 6 CONCLUSION

Using the integral representation of the log-partition function, the Gibbs posterior admits the computation of upper bounds on the true error based on the training data and for any temperature. These bounds are stable under perturbation in the total-variation and Wasserstein metrics, and can be approximated by Langevin Monte Carlo (LMC) algorithms. However, for realistic experiments, the approximations obtained by these algorithms are coarse and require calibration, which leads to rather tight bounds in the interpolation regime of overparametrized neural networks.

The fact that the calibrated bounds are very tight is, at this point, a purely experimental finding, requiring more theoretical investigation in future work.

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

# APPENDIX

In this appendix, we summarize a glossary of notation, give additional theoretical results and missing proofs, and provide more information on the numerical experiments, as well as additional experimental results.

## A    TABLE OF NOTATION

| Notation | Brief description | Section |
|---|---|---|
| $\mathcal{X}$ | space of data | 2 |
| $\Sigma$ | sigma algebra (events) on $\mathcal{X}$ | 2 |
| $\mu$ | probability of data | 2 |
| $n$ | sample size | 1, 2 |
| $\mathbf{x}$ | generic member $(x_1, ..., x_n) \in \mathcal{X}^n$, training sample | 2 |
| $\mathcal{H}$ | hypothesis space | 2 |
| $\Omega$ | sigma algebra (events) on $\mathcal{H}$ | 2 |
| $\ell$ | $\ell : \mathcal{H} \times \mathcal{X} \to [0, \infty)$ loss function | 2 |
| $f$ | secondary loss function | 3.3 |
| $\mathcal{P}(\mathcal{H})$ | probability measures on $\mathcal{H}$ | 2 |
| $\pi$ | nonnegative a-priori measure on $\mathcal{H}$ | 2 |
| $\sigma$ | width of Gaussian prior | 4 |
| $L(h)$ | $L(h) = \mathbb{E}_{x \sim \mu}[h(x)]$, expected loss of $h \in \mathcal{H}$ | 2 |
| $\hat{L}(h, \mathbf{x})$ | $\hat{L}(h, \mathbf{x}) = (1/n) \sum_{i=1}^{n} \ell(h, x_i)$, empirical loss of $h \in \mathcal{H}$ | 2 |
| $\beta$ | inverse temperature | 1, 2 |
| $Z_\beta(\mathbf{x})$ | partition function | 2 |
| $G_{\beta,\pi}(\mathbf{x})$ | Gibbs posterior with energy $\hat{L}$ and prior $\pi$ | 2 |
| $\mathbb{E}_{g \sim G_\beta(\mathbf{x})}$ | posterior expectation | 2 |
| $\boldsymbol{\beta}$ | increasing sequence $(\beta_1 < ... < \beta_K)$ of positive reals | 3.1 |
| $\Gamma(h, \mathbf{x}, \boldsymbol{\beta})$ | bounding functional | 3.1 |
| $F(h, \mathbf{x})$ | placeholder for generalization gap | 3.2 |
| $\kappa$ | $kl(p, q) = p \ln \frac{p}{q} + (1-p) \ln \frac{1-p}{1-q}$, rel. entropy of Bernoulli variables | 2 |
| $\mathrm{KL}(\rho, \nu)$ | $\int \left( \ln \frac{d\rho}{d\nu} \right) d\rho$, KL-divergence of $\rho, \nu \in \mathcal{P}(\mathcal{H})$ | 2, 4.1, 4.2 |
| $d_{TV}(\rho, \nu)$ | total variation distance | 2, 4.1, 4.2 |
| $W_p(\rho, \nu)$ | $p$-Wasserstein distance | 2, 4.1, 4.2 |
| $\Gamma_\nu(h, \mathbf{x}, \boldsymbol{\beta})$ | LMC approximation of $\Gamma(h, \mathbf{x}, \boldsymbol{\beta})$ | 4.2 |
| $\eta$ | step size or learning rate | 4.1 |
| $\nu_{\beta,\eta}$ | invariant measure of LMC approximation of $G_\beta$ with step size $\eta$ | 4.1 |
| $\nu_{\beta,\eta,t}$ | LMC approximation of $G_\beta$ with step size $\eta$ at iteration $t$ | 4.1 |
| $r(\mathbf{x})$ | calibration factor | 5.3, B.4 |
| $\tilde{\mathbf{x}}$ | randomly labeled data | 5.3, B.4 |
| $\mathbb{M}_{\mathrm{stop}}, \mathbb{M}_{\mathrm{erg}}$ | filters for stopping and ergodic mean | 5.2, C.1.3 |

## B    ADDITIONAL RESULTS AND PROOFS

### B.1    SUB-GAUSSIAN LOSSES

The freedom in the choice of $F$ allows a number of bounds to be derived from Theorem 3.2. A centered real random variable $Y$ is called $\sigma$-sub-Gaussian if $\ln \mathbb{E}e^{\lambda Y - \mathbb{E}Y} \leq \lambda^2 \sigma^2 / 2$ for all $\lambda \in \mathbb{R}$. Now, suppose that for some real function $f$ all the $x \in \mathcal{X} \mapsto f(h, x)$ are $\sigma$-sub-Gaussian as $x \sim \mu$. Let $\hat{L}(h, \mathbf{x}) = \frac{1}{n} \sum_{i=1}^{n} f(h, x_i)$ and $L(h) = E_x[f(h, \mathbf{x})]$. Then, $\mathbf{x} \in \mathcal{X}^n \mapsto \hat{L}(h, \mathbf{x})$ as $\mathbf{x} \sim \mu^n$ is $\sigma/\sqrt{n}$-sub-Gaussian. It is tempting to set $F(h, \mathbf{x}) = \lambda \left( L(h) - \hat{L}(h, \mathbf{x}) \right)$ in Theorem 3.2, divide by $\lambda$ and then optimize over $\lambda$. Unfortunately, the last step is impossible, since the optimal $\lambda$ is data-dependent in its dependence on $\Gamma$ and ruins the exponential moment bound on $F$. A more careful argument establishes the following.

**Corollary B.1.** *Suppose that for all $h \in \mathcal{H}$ the random variables $x \in \mathcal{X} \mapsto f(h, x)$ as $x \sim \mu$ are $\sigma$-sub-Gaussian. For $\delta > 0$ with probability at least $1 - \delta$ as $\mathbf{x} \sim \mu^n$ and $h \sim G_{\beta,\pi}(\mathbf{x})$ if $\Gamma(h, \mathbf{x}, \boldsymbol{\beta}) \geq 1$, then,*

$$E_x[f(h, x)] - \frac{1}{n} \sum_i f(h, x_i) \leq \sigma \sqrt{\frac{2[\Gamma(h, \mathbf{x}, \beta)(1 + 1/n) + \ln(\Gamma(h, \mathbf{x}, \boldsymbol{\beta})(n+1)/\delta)]}{n}}.$$

For the proof, we use the following auxiliary result.

**Lemma B.2.** *(Anthony & Bartlett, 1999, Lemma 15.6) Suppose $\Pr$ is a probability distribution and*

$$\{E(\alpha_1, \alpha_2, \delta) : 0 < \alpha_1, \alpha_2, \delta \leq 1\}$$

*is a set of events, such that*

*(i) For all $0 < \alpha \leq 1$ and $0 < \delta \leq 1$,*

$$\Pr\{E(\alpha, \alpha, \delta)\} \leq \delta.$$

*(ii) For all $0 < \alpha_1 \leq \alpha \leq \alpha_2 \leq 1$ and $0 < \delta_1 \leq \delta \leq 1$*

$$E(\alpha_1, \alpha_2, \delta_1) \subseteq E(\alpha, \alpha, \delta).$$

*Then for $0 < a, \delta < 1$,*

$$\Pr \bigcup_{\alpha \in (0,1]} E(\alpha a, \alpha, \delta\alpha(1 - a)) \leq \delta.$$

We put this lemma in a more convenient form.

**Lemma B.3.** *Let $Y$ and $X \geq 0$ be real random variables, $\psi : \mathbb{R} \times (0, 1) \to \mathbb{R}$ be increasing in the 1st argument and $\forall C > 1$, $\delta \in (0, 1)$,*

$$\Pr\{X \leq C \wedge Y > \psi(C, \delta)\} < \delta.$$

*Then, for every $\epsilon > 0$*

$$\Pr\left\{X \geq 1 \wedge Y > \psi\left(X(1 + \epsilon), \frac{\delta\epsilon}{X(1 + \epsilon)}\right)\right\}.$$

*Proof.* This follows from Lemma B.2 using the events

$$E(\alpha_1, \alpha_2, \delta) = \left\{X \leq \alpha_2^{-1} \wedge Y > f\left(\alpha_1^{-1}, \delta\right)\right\}$$

and $a = 1/(1 + \epsilon)$. □

*Proof of Corollary B.1.* Take $F = \lambda\left(L(h) - \hat{L}(h, \mathbf{x})\right)$, so $\ln \mathbb{E}_{\mathbf{x}} \mathbb{E}_{g \sim \pi}\left[e^{F(g, \mathbf{x})}\right] \leq \lambda^2 \sigma^2 / (2n)$. From Theorem 3.2 and the properties of sub-Gaussian variables we get with $\lambda = \sigma^{-1}\sqrt{2n(C + \ln(1/\delta))}$ that

$$\Pr\left\{\Gamma(h, \mathbf{x}, \boldsymbol{\beta}) \leq C \wedge L(h) - \hat{L}(h, \mathbf{x}) > \sigma\sqrt{\frac{2(C + \ln(1/\delta))}{n}}\right\}$$

$$= \Pr\left\{\Gamma(h, \mathbf{x}, \boldsymbol{\beta}) \leq C \wedge L(h) - \hat{L}(h, \mathbf{x}) > \frac{C + \ln(1/\delta)}{\lambda} + \frac{\lambda\sigma^2}{2n}\right\}$$

$$= \Pr\left\{\Gamma(h, \mathbf{x}, \boldsymbol{\beta}) \leq C \wedge \lambda\left(L(h) - \hat{L}(h, \mathbf{x})\right) > C + \lambda^2\sigma^2/(2n) + \ln(1/\delta)\right\} \leq \delta.$$

Substitution in Lemma B.3 with $\psi(C, \delta) = \sigma\sqrt{2(C + \ln(1/\delta))/n}$, $X = \Gamma(h, \mathbf{x}, \boldsymbol{\beta})$, $Y = L(h) - \hat{L}(h, \mathbf{x})$ and $\epsilon = 1/n$ gives Corollary B.1. □

## B.2 PROOFS FOR SECTION 4.2

Restatement of Theorem 4.3:

**Theorem B.4.** *Suppose that $\mathcal{H} = \mathbb{R}^d$ and that there are numbers $m, M < \infty$ such that for every $\mathbf{x}$ in $\mathcal{X}^n$ and $h \in \mathcal{H}$ we have $|\ell(h, \mathbf{x})| \leq m$ and $|F(h, \mathbf{x})| \leq M$. Let $0 = \beta_0 < \beta_1 < \cdots < \beta_K = \beta$ and $\nu_{\beta_k}(\mathbf{x}) \in \mathcal{P}(\mathcal{H})$ be such that $d_{TV}(\nu_{\beta_k}(\mathbf{x}), G_{\beta_k}(\mathbf{x})) = \epsilon_{\beta_k}$. Then,*

*(i) with probability at least $1 - \delta$ as $\mathbf{x} \sim \mu^n$ and $h \sim \nu_\beta(\mathbf{x})$*

$$
\begin{aligned}
F(h, \mathbf{x}) \leq &\ \Gamma_\nu(h, \mathbf{x}, \beta) + \ln \mathbb{E}_{\mathbf{x}} \mathbb{E}_{h \sim \pi}\left[e^{F(h, \mathbf{x})}\right] + \ln \frac{1}{\delta} \\
&+ \ln\left(2 e^{M + \beta m} \epsilon_\beta\right) + \sum_{k=1}^{K}(\beta_k - \beta_{k-1}) m \epsilon_{\beta_{k-1}}.
\end{aligned}
$$

*(ii) with probability at least $1 - \delta$ as $\mathbf{x} \sim \mu^n$*

$$
\begin{aligned}
\mathbb{E}_{h \sim \nu_\beta(\mathbf{x})}[F(h, \mathbf{x})] \leq &\ \mathbb{E}_{h \sim \nu_\beta(\mathbf{x})}[\Gamma_\nu(h, \mathbf{x}, \boldsymbol{\beta})] + \ln \mathbb{E}_{\mathbf{x}} \mathbb{E}_{h \sim \pi}\left[e^{F(h, \mathbf{x})}\right] + \ln \frac{1}{\delta} \\
&+ (M + \beta m)\epsilon_\beta + \sum_{k=1}^{K}(\beta_k - \beta_{k-1}) m \epsilon_{\beta_{k-1}}.
\end{aligned}
$$

*Proof.* By the bound on $\ell$ we have

$$
\Gamma(h, \mathbf{x}, \boldsymbol{\beta}) \leq \Gamma_\nu(h, \mathbf{x}, \boldsymbol{\beta}) + \sum_{k=1}^{K}(\beta_k - \beta_{k-1}) m \epsilon_{\beta_{k-1}}. \tag{9}
$$

(i) From Markov's inequality we have (in analogy to the proof of Theorem 3.2) with probability at least $1 - \delta$ as $x \sim \mu^n$ and $h \sim \nu_\beta(\mathbf{x})$, that

$$
\begin{aligned}
&F(h, \mathbf{x}) + \beta \hat{L}(h, \mathbf{x}) + \ln Z_\beta(\mathbf{x}) \\
\leq &\ \ln \mathbb{E}_{\mathbf{x}} \mathbb{E}_{h \sim \nu_\beta(\mathbf{x})}\left[e^{F(h, \mathbf{x}) + \beta \hat{L}(h, \mathbf{x}) + \ln Z_\beta(\mathbf{x})}\right] + \ln(1/\delta) \\
\leq &\ \ln\left(\mathbb{E}_{\mathbf{x}} \mathbb{E}_{h \sim G_\beta(\mathbf{x})}\left[e^{F(h, \mathbf{x}) + \beta \hat{L}(h, \mathbf{x}) + \ln Z_\beta(\mathbf{x})}\right] + e^{M + \beta m} \epsilon_\beta\right) + \ln(1/\delta) \\
= &\ \ln\left(\mathbb{E}_{\mathbf{x}} \mathbb{E}_{h \sim \pi}\left[e^{F(h, \mathbf{x})}\right] + e^{M + \beta m} \epsilon_\beta\right) + \ln(1/\delta) \\
\leq &\ \ln \mathbb{E}_{\mathbf{x}} \mathbb{E}_{h \sim \pi}\left[e^{F(h, \mathbf{x})}\right] + \ln\left(2 e^{M + \beta m} \epsilon_\beta\right) + \ln(1/\delta).
\end{aligned}
$$

In the second inequality we used $\ln Z_\beta(\mathbf{x}) \leq 0$ and in the last line we used for $a, b \geq 1$ that $\ln(a + b) \leq \ln \max\{a, b\} + \ln 2 \leq \ln a + \ln b + \ln 2 = \ln a + \ln 2b$. Subtract $\beta \hat{L}(h, \mathbf{x}) + \ln Z_\beta(\mathbf{x})$, use (4) and (9).

(ii) Again with Markov's inequality, with probability at least $1 - \delta$ as $x \sim \mu^n$

$$
\begin{aligned}
&\mathbb{E}_{h \sim \nu_\beta(\mathbf{x})}\left[F(h, \mathbf{x}) + \beta \hat{L}(h, \mathbf{x}) + \ln Z_\beta(\mathbf{x})\right] \\
\leq &\ \ln \mathbb{E}_{\mathbf{x} \sim \mu^n}\left[e^{\mathbb{E}_{h \sim \nu_\beta(\mathbf{x})}[F(h, \mathbf{x}) + \beta \hat{L}(h, \mathbf{x}) + \ln Z_\beta(\mathbf{x})]}\right] \\
\leq &\ \ln \mathbb{E}_{\mathbf{x} \sim \mu^n}\left[e^{\mathbb{E}_{h \sim G_\beta(\mathbf{x})}[F(h, \mathbf{x}) + \beta \hat{L}(h, \mathbf{x}) + \ln Z_\beta(\mathbf{x})] + (M + \beta m)\epsilon_\beta}\right].
\end{aligned}
$$

In the second inequality we used the fact that $\ln Z_\beta(\mathbf{x}) \leq 0$ and the bounds on $F$ and $\ell$. Then, Jensen's inequality bounds the last line as

$$
\begin{aligned}
&\ln \mathbb{E}_{\mathbf{x} \sim \mu^n} \mathbb{E}_{h \sim G_\beta(\mathbf{x})}\left[e^{F(h, \mathbf{x}) + \beta \hat{L}(h, \mathbf{x}) + \ln Z_\beta(\mathbf{x})}\right] e^{(M + \beta m)\epsilon_\beta} \\
= &\ \ln \mathbb{E}_{\mathbf{x} \sim \mu^n} \mathbb{E}_{h \sim \pi}\left[e^{F(h, \mathbf{x})}\right] + (M + \beta m)\epsilon_\beta.
\end{aligned}
$$

Again subtract $\beta \hat{L}(h, \mathbf{x}) + \ln Z_\beta(\mathbf{x})$, use (4) and (9). $\qquad\square$

### B.3 MISCELLANEOUS LEMMATA

**Lemma B.5.** *For* $0 < \beta < \infty$

$$\max \left\{ KL\left(G_\beta, G_{2\beta}\right), KL\left(G_{2\beta}, G_2\right) \right\} \leq \beta \left( \mathbb{E}_{h \sim G_\beta} \left[ \hat{L}\left(h\right) \right] - \mathbb{E}_{h \sim G_{2\beta}} \left[ \hat{L}\left(h\right) \right] \right)$$

*Proof.* Using Lemma 3.1

$$
\begin{aligned}
\mathrm{KL}\left(G_\beta, G_{2\beta}\right) &= \mathbb{E}_{h \sim G_\beta} \left[ -\beta \hat{L}\left(h\right) - \ln Z_\beta + 2\beta \hat{L}\left(h\right) + \ln Z_{2\beta} \right] \\
&= \mathbb{E}_{h \sim G_\beta} \left[ \beta \hat{L}\left(h\right) \right] - \int_\beta^{2\beta} \mathbb{E}_{h \sim G_\gamma} \left[ \hat{L}\left(h\right) \right] d\gamma \\
&\leq \beta \left( \mathbb{E}_{h \sim G_\beta} \left[ \hat{L}\left(h\right) \right] - \mathbb{E}_{h \sim G_{2\beta}} \left[ \hat{L}\left(h\right) \right] \right).
\end{aligned}
$$

Similarly

$$
\begin{aligned}
\mathrm{KL}\left(G_{2\beta}, G_2\right) &= \mathbb{E}_{h \sim G_{2\beta}} \left[ -2\beta \hat{L}\left(h\right) - \ln Z_{2\beta} + \beta \hat{L}\left(h\right) + \ln Z_\beta \right] \\
&= -\mathbb{E}_{h \sim G_\beta} \left[ \beta \hat{L}\left(h\right) \right] + \int_\beta^{2\beta} \mathbb{E}_{h \sim G_\gamma} \left[ \hat{L}\left(h\right) \right] d\gamma \\
&\leq \beta \left( \mathbb{E}_{h \sim G_\beta} \left[ \hat{L}\left(h\right) \right] - \mathbb{E}_{h \sim G_{2\beta}} \left[ \hat{L}\left(h\right) \right] \right).
\end{aligned}
$$

$\square$

### B.4 THE CALIBRATION FACTOR

We assume that the computed functional $\Gamma_\nu\left(h, \mathbf{x}, \beta\right)$ fails to estimate $\Gamma\left(h, \mathbf{x}, \beta\right)$ by a factor $r\left(\mathbf{x}\right) > 0$, which we compute as

$$r\left(\mathbf{x}\right) = \min \left\{ r : \forall k \in [K], \ \kappa^{-1} \left( \mathbb{E}_{h \sim \nu_{\beta_k}(\mathbf{x})} \left[ \hat{L}_{01}\left(h, \tilde{\mathbf{x}}\right) \right], \frac{1}{n} \left( r \Gamma_\nu\left(h, \tilde{\mathbf{x}}, \beta_1^k\right) + \ln \frac{2\sqrt{n}}{\delta} \right) \right) \geq \frac{1}{2} \right\},$$

where $\tilde{\mathbf{x}}$ is the training set $\mathbf{x}$ with random labels and $\hat{L}_{01}$ the empirical 01-error. The calibration value $r$ is thus the smallest factor of $\Gamma_\nu$, for which we obtain a correct upper bound on the 01-error with random labels for all the $\beta_k$.
We emphasize that the calibration procedure depends only on the training data.

## C EXPERIMENTAL DETAILS AND ADDITIONAL RESULTS

### C.1 EXPERIMENTAL DETAILS

All the codes to reproduce the results are provided through this `https://anonymous.4open.science/r/Gibbs-Generalization-45F1`. For all the experiments we use an isotropic Gaussian prior with $\mu = 0$, for bounded loss with $\sigma = 5$ and for unbounded loss with $\sigma = 0.1$. This induces an L2-regularization term in the energy function that is stated in the proof of Corollary 4.2. The confidence parameter $\delta$ appearing in our bounds is set to $0.01$ for all experiments

We use either standard SGLD or ULA with a constant step size and without additional correction terms. When ULA has been used, we use a step size of 0.01 for both datasets. However with SGLD, we set the step size to 0.01 for MNIST and 0.005 for CIFAR-10. For both datasets, MNIST and CIFAR-10, we use neural networks with ReLU activation functions.

### C.1.1 NETWORK ARCHITECTURE

The fully connected networks consist of one, two, three hidden layers, each containing a constant number of units. Besides that we are using LeNet-5 architecture for MNIST and VGG16 architecture for CIFAR-10 to achieve low test error. For loss function $\ell$, we are mostly using bounded loss function such as bounded binary cross-entropy (BBCE) as described in Appendix D of Dziugaite & Roy (2018) or the Savage loss (Masnadi-Shirazi & Vasconcelos, 2008). As unbounded loss function we tried binary cross-entropy (BCE) (Section C.2.7), but with a smaller value of $\sigma$, so as to avoid excessive training errors for small values of $\beta$.

The LeNet-5 network follows a systematic pattern of alternating convolutional and pooling layers, followed by fully connected layers (LeCun et al., 2002). It begins with an input layer that accepts $32 \times 32$ grayscale images. Thus, we pad our images to fit. The first convolutional layer (C1) applies 6 filters of size $5 \times 5$ to extract low-level features, followed by a $2 \times 2$ average pooling layer (S2) for spatial downsampling. The second convolutional layer (C3) uses 16 filters of size $5 \times 5$ to capture more complex feature combinations, followed again by a $2 \times 2$ average pooling layer (S4). A third convolutional layer (C5) with 120 filters of size $5 \times 5$ acts as a feature extractor, producing 120 feature maps, each of size $1 \times 1$. The architecture concludes with two fully connected layers: F6 with 84 neurons and a final output layer with 10 neurons for the original digit classification task. However, for our binary classification task, we modify F6 to have 420 neurons and use a single-neuron output layer. Throughout the network, ReLU activation functions replace the original tanh activations, which improves gradient flow and training performance in modern implementations.

VGG-16 is a widely used deep convolutional neural network architecture known for its simplicity and strong performance in image classification tasks (Simonyan & Zisserman, 2014). The architecture follows a consistent design using only $3 \times 3$ convolutional filters and $2 \times 2$ max pooling operations throughout the network. In our implementation, VGG-16 is adapted to handle CIFAR-10's smaller $32 \times 32$ RGB images. The network consists of 13 convolutional layers organized into five blocks: the first two blocks contain two convolutional layers each with 64 and 128 filters, respectively, while the last three blocks contain three convolutional layers each with 256, 512, and 512 filters, respectively. Each block is followed by a $2 \times 2$ max pooling layer for spatial downsampling. All convolutional layers employ $3 \times 3$ kernels with padding to preserve spatial dimensions, and ReLU activation functions introduce non-linearity. The convolutional feature extractor is followed by a classifier head consisting of three fully connected layers: two hidden layers with 1024 neurons each, using ReLU activation, and a final output layer with 1 neurons for binary classification. We also removed dropout to ensure that SGLD minimizes the defined energy function without any additional terms.

For MNIST, the input is a 784-dimensional vector, and the output is a scalar since we perform binary classification between digits 0–4 and 5–9. For CIFAR-10, the input dimension is 3072, and the output is again scalar, corresponding to binary classification between vehicles and animals. For evaluating our models, we are using all 10,000 test examples for both datasets.

### C.1.2 MINIBATCHES

When using SGLD, we adopt minibatches of size proportional to $\sqrt{n}$. Thus, for $n = 2000$ the mini-batch size is 50, and for $n = 8000$ it is 100.

### C.1.3 MOVING AVERAGE FILTERS

As we explained in Section 5.2, we are using a running mean $\mathbb{M}(x_1, \cdots, x_t)$ of $\hat{L}(h_j, \mathbf{x})$ from $j = 1, \cdots, t$ both as a criterion to stop the experiment and an estimation for $\mathbb{E}_{h \sim G_{\beta_k}} \left[ \hat{L}(h, \mathbf{x}) \right]$. We define the running mean recursively in one of two ways:

$$\mathbb{M}_t = \tfrac{\alpha}{2}\hat{L}(h_t, \mathbf{x}) + \tfrac{\alpha}{2}\hat{L}(h_{t-1}, \mathbf{x}) + (1 - \alpha)\mathbb{M}_{t-1},$$
$$\mathbb{M}_t = \alpha\hat{L}(h_t, \mathbf{x}) + (1 - \alpha)\mathbb{M}_{t-1},$$

with $\mathbb{M}_0 = 1$ and small $\alpha$. We use the first (symmetric) form in the experiments with ULA, and the second (standard exponential moving average) form with SGLD for convenience. We set different values of $\alpha$ for the two roles: $\alpha = 0.0025$ for the stopping criterion ($\mathbb{M}_{\text{stop}}$) and $\alpha = 0.01$ for

approximating the ergodic mean ($\mathbb{M}_{\mathrm{erg}}$). The stopping rule is triggered when

$$\mathbb{M}_t - \mathbb{M}_{t-1} \geq \epsilon,$$

with $\epsilon = 10^{-7}$. To avoid premature termination, we impose a minimum of 4000 steps before applying this criterion. As $\alpha \to 0$ and $t \to \infty$, the quantity $\mathbb{M}_t$ converges to the ergodic mean.

## C.2 EXPERIMENTAL RESULTS

### C.2.1 ILLUSTRATION OF BOUND COMPUTATION

In this section, we demonstrate again the figure in the main body in more details. The figure 2 illustrates how our bounds are computed. The sequence of mean training losses in $\ell$ is used to compute for each $\beta$ the functional $\Gamma$ and the "KL-Bound", which corresponds to the right hand side of the inequalities in Corollary 3.3. Our bound on the test loss is then computed by applying the function $\kappa^{-1}$ to the empirical 0-1 error and to this kl-bound. The graph of "KL(Train, Test)" corresponds to the left hand side in Corollary 3.3.

It is remarkable that the close fit of the upper bound on the random labels is achieved by the adjustment of a single calibration parameter.

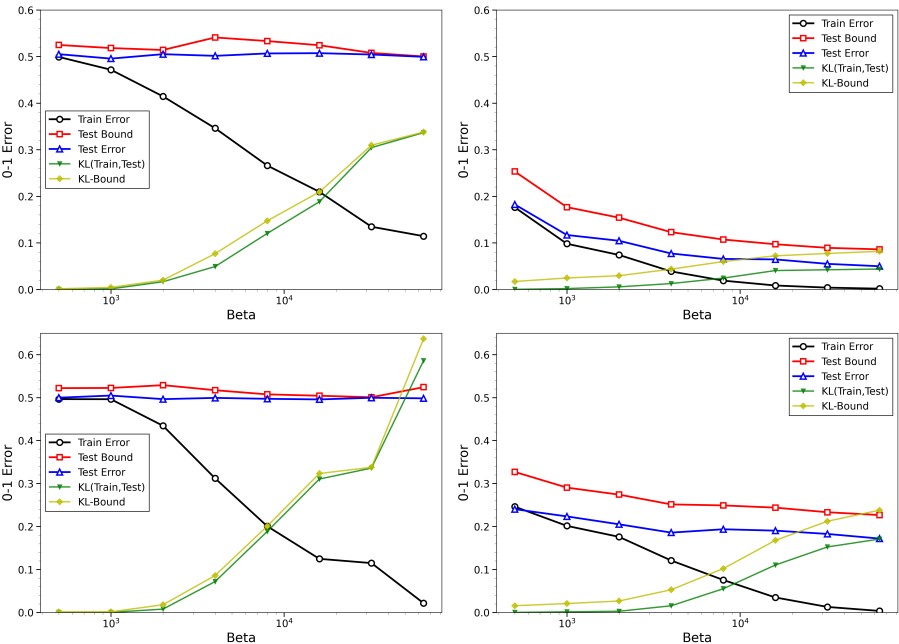

Figure 2: A more detailed version of Figure 1 to illustrate how the bounds are computed.

### C.2.2 SINGLE-DRAWS

For the setting described in Section 5.4, we also present the bounds for the single-draw case in Figure 3. It is noteworthy that, although the theoretical guarantees for this scenario are rather weak, the empirical bounds behave well. However, as visible in the plots, the results exhibit fluctuations and irregularities caused by stochastic effects, which make them less reliable.

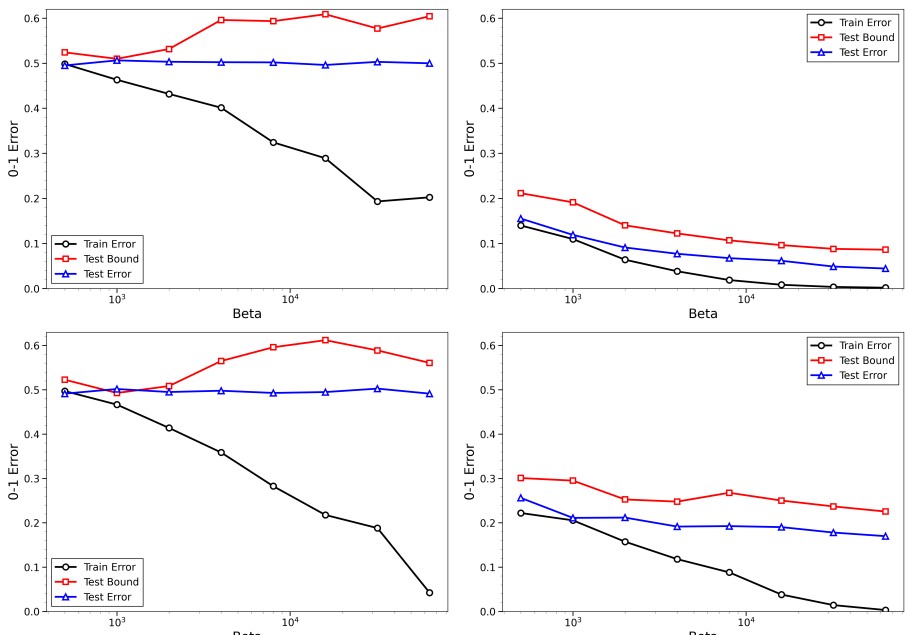

Figure 3: SGLD on MNIST and CIFAR-10 with 8000 training examples using BBCE loss function. The first row corresponds to MNIST and the second row to CIFAR-10. Random labels are shown on the left, correct labels on the right. Both random and true labels are trained with exactly the same algorithm and parameters on a fully connected ReLU network with two hidden layers of 1000 (respectively 1500) units. The calibration factor for MNIST is 0.77, for CIFAR-10 0.89. Train error, test error and our bound for a single-draw of the 0-1 loss are plotted against $\beta$.

### C.2.3 DIFFERENT ARCHITECTURES

In this section, we evaluate the performance of different models and architectures on both MNIST and CIFAR-10, demonstrating that our bound can be used to guide model selection. In addition to the two-hidden-layer neural networks described in Section 1, we consider fully connected neural networks with three hidden layers, containing 500 and 1000 units for MNIST and CIFAR-10, respectively. Furthermore, we employ the LeNet-5 architecture for MNIST and VGG-16 for CIFAR-10 to achieve high test accuracy. Detailed descriptions of these architectures are provided in Section C.1.1.

Figure 4 demonstrates the robustness of our bound across different models. We observe that the bounds can be very tight even when the test error is small. For convolutional neural networks, especially on the MNIST dataset, we observe strong performance with the true labels, but relatively poor performance with random labels, despite having more parameters than training examples. This can be explained by the fact that convolutional architectures are still far from being highly overparameterized. For the MNIST dataset, we use fully connected neural networks with two or three hidden layers, containing 1000 or 500 units per layer, respectively. This corresponds to a total of approximately 1,787,000 and 893,000 parameters, resulting in a parameter-to-training-example ratio of roughly 200 and 100, respectively. In contrast, LeNet-5 has around 100,000 parameters, yielding a ratio of approximately 12.5.

The empirical test bounds can serve as a selection criterion among different models. Table 1 show that test bounds at low temperature are useful for model selection, and that bounds at high temperature can also predict the behavior of the model at low temperature.

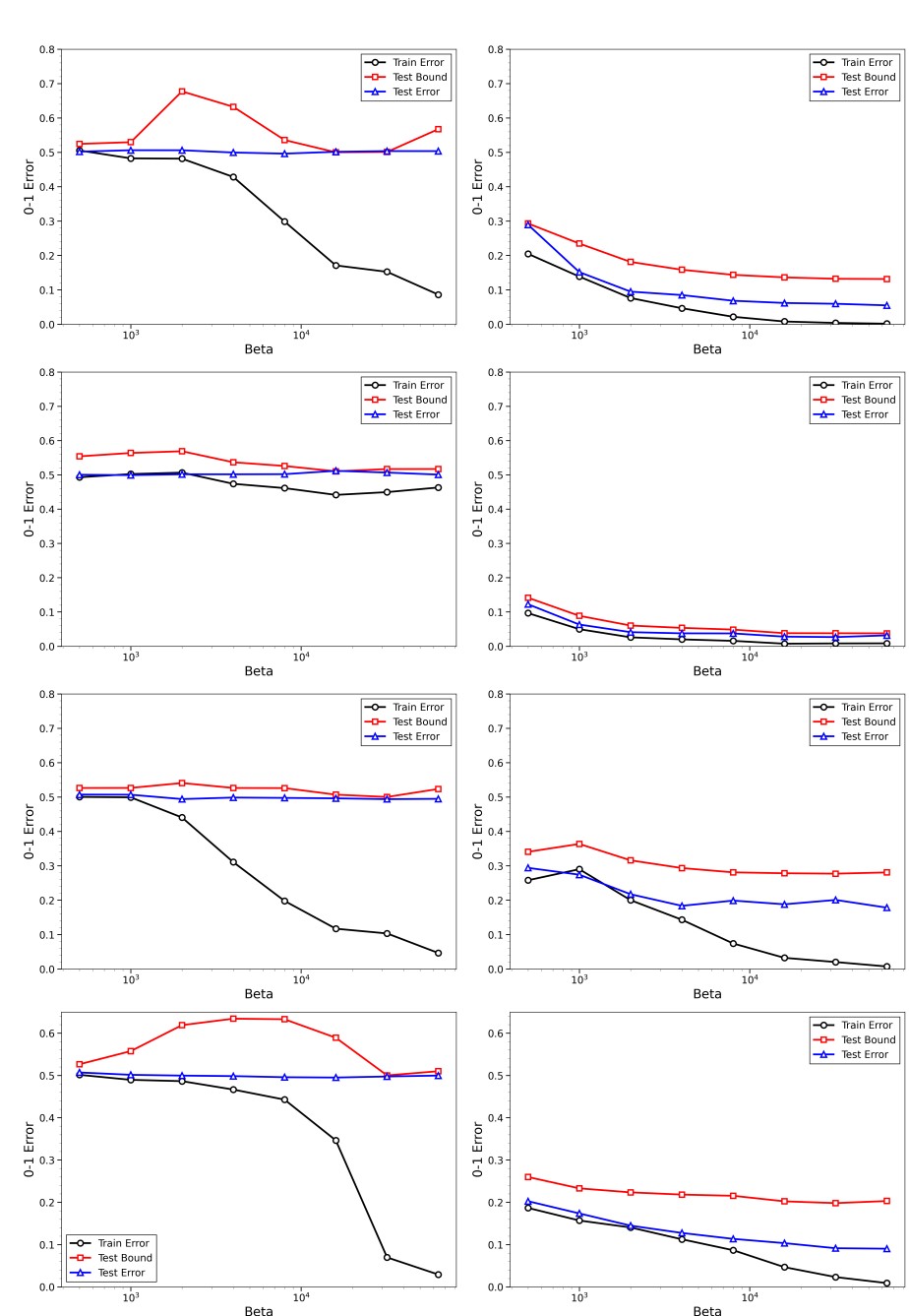

Figure 4: SGLD on MNIST and CIFAR-10 with 8000 training examples using BBCE loss function. The first two rows correspond to MNIST, and the remaining rows to CIFAR-10. Random labels are shown on the left, and correct labels on the right. Both random and true labels are trained using the same algorithm and hyperparameters on a fully connected ReLU network with three hidden layers of 500 (MNIST) or 1000 (CIFAR-10) units, followed by LeNet-5 (MNIST) or VGG-16 (CIFAR-10) shown in the subsequent row. The calibration factors for MNIST are 0.26 and 0.08, for CIFAR-10 0.24 and 0.18. The training error, test error, and our bound for the Gibbs posterior average of the 0–1 loss are plotted against $\beta$.

|                              | 2HL (W=1000) | 3HL (W=500) | LeNet-5 |
|------------------------------|--------------|-------------|---------|
| Test Bound at $\beta = 1k$   | 0.1766       | 0.2347      | 0.0887  |
| Test Error at $\beta = 64k$  | 0.0498       | 0.0549      | 0.0317  |
| Test Bound at $\beta = 64k$  | 0.0860       | 0.1314      | 0.0375  |

(a) MNIST, 8k training examples (true labels).

|                              | 2HL (W=1500) | 3HL (W=1000) | VGG-16 |
|------------------------------|--------------|--------------|--------|
| Test Bound at $\beta = 1k$   | 0.2905       | 0.3635       | 0.2330 |
| Test Error at $\beta = 64k$  | 0.1719       | 0.1782       | 0.0903 |
| Test Bound at $\beta = 64k$  | 0.2266       | 0.2807       | 0.2030 |

(b) CIFAR-10, 8k training examples (true labels).

Table 1: Test bounds and test errors for different neural network architectures on MNIST and CIFAR-10. The bounds at both low and high temperatures reliably reflect test error performance at low temperature.

### C.2.4    ULA

We have also conducted experiments using ULA for both datasets. The main difference from SGLD is that we use all the information to compute the gradient at each step. The results are shown in Figure 5.

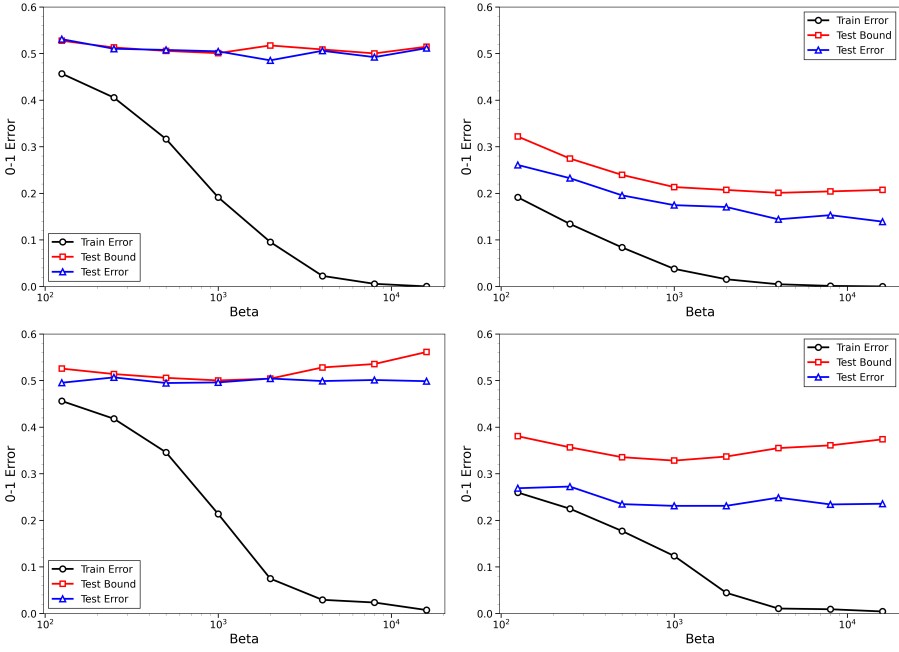

Figure 5: ULA on MNIST and CIFAR-10 with 2000 training examples using BBCE loss function. The first row corresponds to MNIST and the second row to CIFAR-10. Random labels are shown on the left, correct labels on the right. Both random and true labels are trained with the same algorithm and parameters on a fully connected ReLU network with one (respectively two) hidden layers of 500 (respectively 1000) units. The calibration factor for MNIST is 0.49, for CIFAR-10 0.46. Train error, test error and our bound for the Gibbs posterior average of the 0-1 loss are plotted against $\beta$.

### C.2.5 SAVAGE LOSS FUNCTION

We additionally performed experiments using the Savage loss to verify the robustness of our results across different loss functions. Following the same setup as in the previous section, the outcomes are reported in Figure 7.

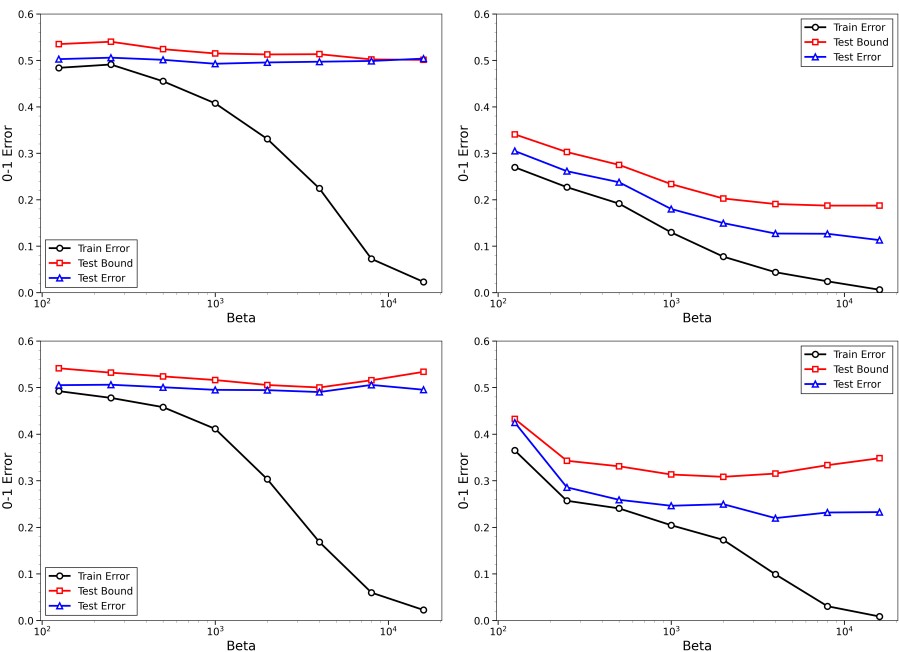

Figure 6: ULA on MNIST and CIFAR-10 with 2000 training examples using Savage loss function. The first row corresponds to MNIST and the second row to CIFAR-10. Random labels are shown on the left, correct labels on the right. Both random and true labels are trained with the same algorithm and parameters on a fully connected ReLU network with one (respectively two) hidden layers of 500 (respectively 1000) units. The calibration factor for MNIST is 0.49, for CIFAR-10 0.59. Train error, test error and our bound for the Gibbs posterior average of the 0-1 loss are plotted against $\beta$.

### C.2.6 UNCALIBRATED BOUNDS

We show uncalibrated bounds for the MNIST dataset with the BBCE and Savage loss functions under the several experimental conditions including setups of Sections C.2.4 and C.2.5. The bounds are somewhat looser than the calibrated ones, but still far from trivial. As in all other scenarios the test errors are upper bounded correctly.

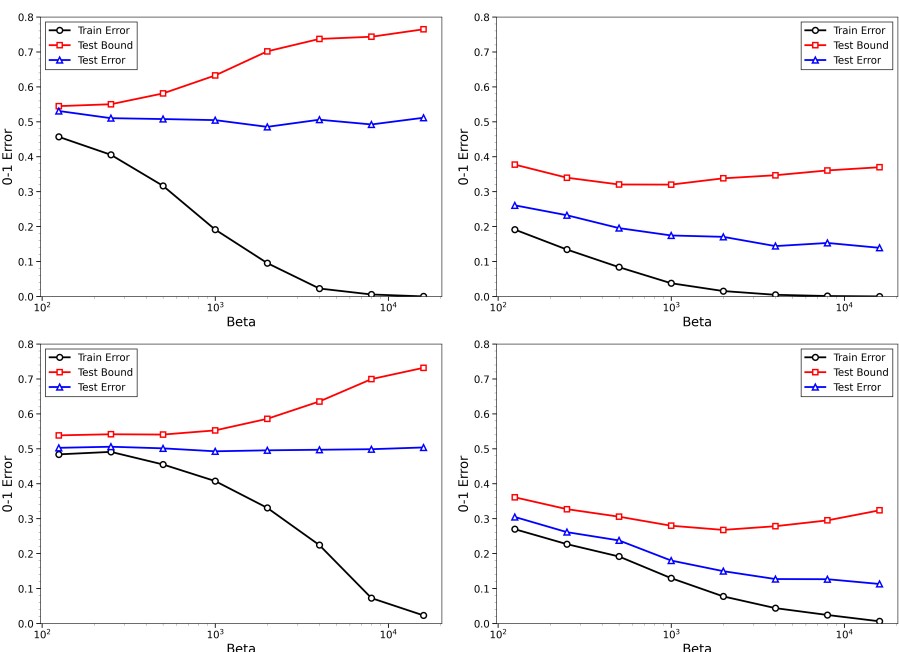

Figure 7: ULA on MNIST with 2000 training examples using BBCE and Savage loss functions. The first row corresponds to MNIST with BBCE and the second row to Savage. Random labels are shown on the left, correct labels on the right. Both random and true labels are trained with the same algorithm and parameters on a fully connected ReLU network with one hidden layers of 500 units. Train error, test error and our bound for the Gibbs posterior average of the 0-1 loss are plotted against $\beta$ without any calibration.

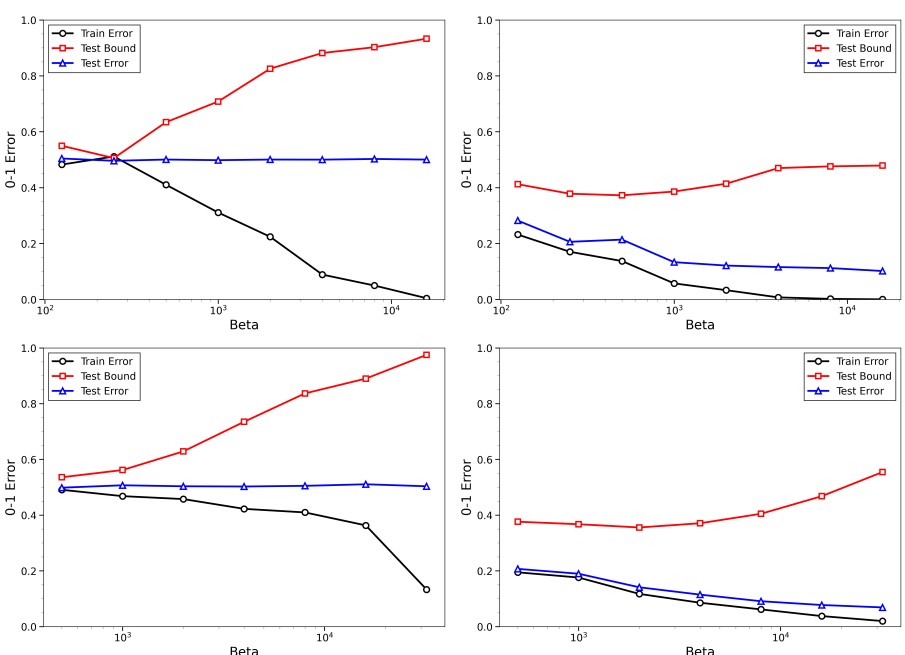

Figure 8: SGLD/ULA on MNIST with 2000/8000 training examples using BBCE/Savage loss functions. The first row corresponds to MNIST with 2000 samples with SGLD and BBCE and the second row to 8000 samples with ULA and Savage. Random labels are shown on the left, correct labels on the right. Both random and true labels are trained with the same algorithm and parameters on a fully connected ReLU network with two hidden layers of 1000 units and one hidden layer of 500 units respectively. Train error, test error and our bound for the Gibbs posterior average of the 0-1 loss are plotted against $\beta$ without any calibration.

### C.2.7 UNBOUNDED LOSS FUNCTION

In this section, we use the binary cross-entropy loss to compute the $\Gamma$ functional. Since binary cross-entropy is unbounded, the loss can become very large at high temperatures. To avoid this issue, we set the standard deviation of the Gaussian prior to 0.1 in this section. The following plot shows the results under the same setup as Section C.2.5, except that we use binary cross-entropy instead of the Savage loss.

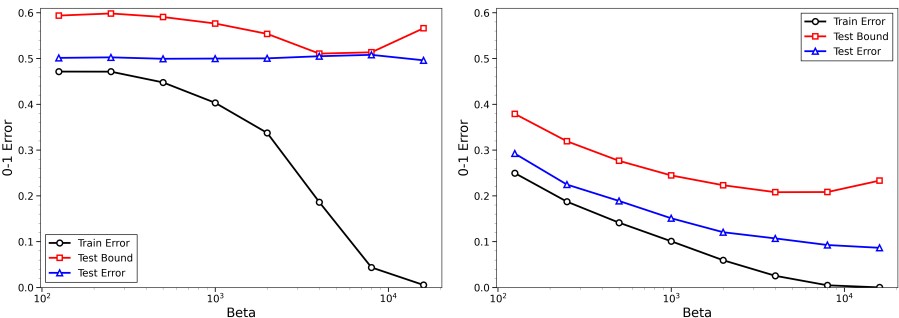

Figure 9: ULA on MNIST with 2000 training examples using binary cross entropy loss function. Random labels are shown on the left, correct labels on the right. Both random and true labels are trained with the same algorithm and parameters on a fully connected ReLU network with one hidden layers of 500 units. The calibration factor is 0.34. Train error, test error and our bound for the Gibbs posterior average of the 0-1 loss are plotted against $\beta$.

### C.2.8 REAL-WORLD USE CASES

We further evaluated Stochastic Gradient Descent (SGD) to examine the practical relevance of our bounds in real-world interpolation regimes.

Based on our observations, we suggest the following procedure for practitioners who wish to train overparameterized neural networks with standard SGD while also obtaining generalization guarantees. First, randomly permute the labels, train the network at different temperatures, and compute the bound together with the calibration factor. Then, repeat the same procedure using the true labels. At very low temperatures, this approach provides generalization guarantees that may transfer to SGD. The corresponding results are presented in Table 2.

|  | 2HL (W=1000) | 3HL (W=500) | LeNet-5 |
|---|---|---|---|
| Test Error, SGD | 0.0364 | 0.0363 | 0.0308 |
| Test Error, SGLD ($\beta = 64k$) | 0.0498 | 0.0549 | 0.0317 |
| Test Bound, SGLD ($\beta = 64k$) | 0.0860 | 0.1314 | 0.0375 |

(a) MNIST, 8k training examples (true labels).

|  | 2HL (W=1500) | 3HL (W=1000) | VGG-16 |
|---|---|---|---|
| Test Error, SGD | 0.1423 | 0.1415 | 0.0933 |
| Test Error, SGLD ($\beta = 64k$) | 0.1719 | 0.1782 | 0.0903 |
| Test Bound, SGLD ($\beta = 64k$) | 0.2266 | 0.2807 | 0.2030 |

(b) CIFAR-10, 8k training examples (true labels).

Table 2: Comparing SGD test error with SGLD test errors and bounds for different neural network architectures on MNIST and CIFAR-10.

