# OpenReview forum: "Generalization of Gibbs and Langevin Monte Carlo Algorithms in the Interpolation Regime"
_ICLR.cc/2026/Conference — Submitted to ICLR 2026_

### Official Review · Reviewer_FGWz · 2025-10-20

**Soundness:** 3
**Presentation:** 3
**Contribution:** 2
**Rating:** 2
**Confidence:** 3

**Summary:**

The paper introduces a PAC-Bayesian bound based generalization error. To make the bound applicable in practice, it is calibrated on the data with randomized labels. The bound is then tested on benchmark problems.

**Strengths:**

The paper is very transparent in the assumptions it makes and the conclusions it draws. I appreciate appendix C2 which provides experiments with different architectures and training losses.

**Weaknesses:**

It is disappointing that the bound needs to be calibrated to hold in practice. To estimate the bound, one needs to repeat the experiment with randomized labels and for different temperatures, which is significantly more expensive than directly estimating the test loss. Combining this with the fact that the calibrated bound does not rigorously hold I am unsure of the proposed use the bound.

Given that the left-hand-side of Figure 1 is not a valid test of the bound (it is calibrated to hold there) the set of experiments where the bound is tested is somewhat small.

**Questions:**

1. What is the proposed use of this bound? Could it be used instead of cross validation?
2. The experiments section seems to suggest that the bound is not practically applicable for some losses. Can the authors provide some guidance here?
3. How does the bound accuracy depend on the number of temperature levels?

---

> ### Author Response · Authors · 2025-11-18
> **Official Response by Authors to Reviewer FGWz**
>
> We appreciate the reviewer's evaluation and comments. In what follows, we aim to respond to the reviewer's questions and additional comments.
>
> ### Weaknesses
>
> * **Calibration**: Our bound can also be applied __without__ calibration, as documented in the updated version.  The purpose of calibration is to tighten the bound using training data alone and to account for terms in Theorem 4.3 that cannot be estimated directly. The point of showing the training errors for random labels is to demonstrate that the bound holds
> in the interpolation regime, both for random and true labels.
>
> * **Regarding computational cost**: The computational cost of calibration of the bound is not a bottleneck, as it is only twice the cost without calibration.
>
> > "To estimate the bound, one needs to repeat the experiment with randomized labels and for different temperatures, which is significantly more expensive than directly estimating the test loss."
>
> If you are referring to cross-validation, clearly, estimating the test loss with cross-validation does not explain generalization.
>
> ### Questions and minor issues
> > "1. What is the proposed use of this bound? Could it be used instead of cross-validation?"
>
> Please refer to our answer above.
>
> > "2. The experiments section seems to suggest that the bound is not practically applicable for some losses. Can the authors provide some guidance here?"
>
> Apart from it being nonnegative, Theorem 3.2 makes no assumptions on the loss function. For the Langevin approximation, it needs to be differentiable with a Lipschitz gradient. The updated version also shows experiments with an unbounded loss function.
>
> > "3. How does the bound accuracy depend on the number of temperature levels?"
>
> In the experiment, we always compute the bound with 8 different temperature values. Adding more values would tighten the bounds; however, our results appear to be already satisfactory with 8.

---

> > ### Comment · Reviewer_FGWz · 2025-11-22
> >
> > I thank the authors for the added results, I am glad to see that the bound works even without the calibration, I will somewhat increase the score.
> > However, the cost of calibration is indeed twice the cost without calibration, but there is still the cost of running the experiment at different temperatures, which is quite significant. And it is still not clear to me what is the use of this bound.

---

> ### Author Response · Authors · 2025-11-22
>
> We thank the reviewer for their follow-up and for continuing the discussion.
> ### Utility of the bound
> As highlighted in the introduction, overparameterized neural networks can achieve nearly zero training error even on data with randomized labels, yet still exhibit large test errors (e.g., Figure 1 in Zhang et al., 2016; 2021). Understanding generalization in such settings is a central challenge in machine learning theory.
>
> According to our bound, the **only term** that distinguishes models trained on true labels from those trained on random labels is the $\mathbf{\Gamma}$ **functional**, which depends only on the training data. This suggests that **the training error at high temperature** (i.e., high noise), where optimization is easier, **is predictive of the test error at low temperature**, where optimization is difficult. See Figure 1 of our paper for the different behavior of the training loss for random and true labels. **An application to model selection** is shown in **Table 1** of the paper.

---

> > ### Author Response · Authors · 2025-11-28
> > **follow up on our comments on utility of the bound**
> >
> > We once again thank the reviewer for their feedback. We would like to ask whether our last comments have addressed their questions and if the reviewer has remaining concerns we would be glad to further engage in discussion.
> >
> > Evaluating the bound requires computing quantities at multiple temperature levels. This dependence is inherent in the structure of the bound itself, since generalization is expressed through a temperature-resolved integral.

---

### Official Review · Reviewer_V25C · 2025-10-27

**Soundness:** 3
**Presentation:** 2
**Contribution:** 2
**Rating:** 6
**Confidence:** 3

**Summary:**

This paper proposes new generalization bounds for the Gibbs algorithm, where the algorithm output is sampled from a Gibbs posterior whose potential is proportional to the empirical risk. The main result are based on PAC-Bayesian argument and an integral decomposition of the error across the temperature parameter. This bound suggests that the generalization error at a given temperature is linked to the generalization error at higher temperatures. An important feature of the proposed theory is to be stable when the Gibbs posteriors at various temperatures are approximated (eg, by MCMC algorithms), making the bound computable in practice. Therefore, the theory is supported by experiments on the MNIST and CIFAR10 datasets.

**Strengths:**

- The proof technique explicitly links the generalization error at different temperatures.
- The stability by approximation makes the bounds computable in practice (with potentially an important computational cost).
- The new bounds are data-dependent and fully computable in practice

**Weaknesses:**

*Main weaknesses:*
 - All results are written with placeholder functions in the main text. It would improve the readability to include actual generalization bounds directly in the main text.
 - Due to the representation of the bound as a discrete integral over several higher temperatures, the experiments are very computationally heavy, as Gibbs posteriors at several temperatures have to be approximated. This might diminish the practical reach of the proposed theory.

*Other (more minor) issues:*
 - Line 73: $\epsilon$ should be $\epsilon_{\beta_k}$
 - Line 105, $P(H) \times P(H)$ should be $P(H \times H)$.
 - Lline 206, as $\exp(F)$ is positive, it seems to me that we do not need much condition to exchange the two expectations
 - $KL$ would be more beautifully written $\mathrm{KL}$.

**Questions:**

- Is it correct that Corollary 4.2 is a consequence of known results?
- Line 415: why do we need to distinguish between $\beta$ and $2\beta$
- Do you think that it could be possible to make the experiments less computationally heavy by using the estimation of the posterior at higher temperatures as a kind of "warm-start" for lower temperatures, if that can make sense in your setting?

---

> ### Author Response · Authors · 2025-11-18
> **Official Response by Authors to Reviewer V25C**
>
> We appreciate the reviewer's evaluation and comments. In what follows, we aim to respond to the reviewer's questions and additional comments.
>
> ### Weaknesses
>
> > “All results are written with placeholder functions in the main text. It would improve the readability to include actual generalization bounds directly in the main text.”
>
> We appreciate this suggestion. In the revised version, we have updated Section 3.3 to make the statement of the bounds more explicit. In particular, we have commented about how the bound can be applied to the 01-loss.
>
> > “Due to the representation of the bound as a discrete integral over several higher temperatures, the experiments are very computationally heavy… This might diminish the practical reach of the proposed theory.”
>
> Please see our response to your last question below.
>
> ### Questions and Minor Issues
>
> > “Is it correct that Corollary 4.2 is a consequence of known results?”
>
> Yes, essentially. We have added more details in the revision version to make this connection clearer.
>
> > “Line 415: why do we need to distinguish between β and 2β?”
>
> Since our bound depends on the absolute and relative values of the training error at different temperatures, it is necessary to distinguish them.
>
> > “Could the experiments be made less computationally heavy by using the posterior at higher temperatures as a warm start for lower temperatures?”
>
> It makes a lot of sense, and we plan to implement an annealing procedure, which gradually decreases the temperature to compute the bounds for the entire temperature range in a single sweep. This should significantly decrease the computational cost. This said, the contribution of the paper is mainly on the learning theory side, and the primary focus is not on computational efficiency.
>
> **Minor issues**: We thank the reviewer for carefully noting the typos; all have been corrected in the revision.
> Regarding the comment about $\epsilon$ on line 73, we could not locate this instance and would appreciate any additional clarification.

---

> > ### Comment · Reviewer_V25C · 2025-11-19
> > **Thank you for your answer**
> >
> > Thank you very much for your answer. Regarding the typo line 73, there is indeed a typo on my side, sorry.
> > I think I meant line 373.

---

> > > ### Author Response · Authors · 2025-11-20
> > >
> > > Thank you for the clarification. We appreciate it. The typo at line 373 has already been corrected in the updated version that we uploaded.

---

### Official Review · Reviewer_miYM · 2025-10-31

**Soundness:** 2
**Presentation:** 2
**Contribution:** 2
**Rating:** 4
**Confidence:** 1

**Summary:**

This paper is way out of my expertise, I am not in a position to offer a meaningful evaluation.

**Strengths:**

.

**Weaknesses:**

.

**Questions:**

.

---

> ### Author Response · Authors · 2025-11-18
> **Official Response by Authors to Reviewer miYM**
>
> We thank the reviewer for their time and for honestly noting that the paper falls outside their area of expertise. We appreciate the feedback and have no additional comments.

---

### Official Review · Reviewer_rD1H · 2025-11-01

**Soundness:** 3
**Presentation:** 2
**Contribution:** 1
**Rating:** 2
**Confidence:** 2

**Summary:**

The paper provides a mechanism for bounding test error in overparametrized regimes for Gibbs sampling algorithms (and approximations such as Langevin Monte Carlo), which accurately captures generalization error in the low temperature regime.

**Strengths:**

The paper shows a scheme for bounding the test error which is rigorously derived and implementable.

In the experiments presented, the predictions appear to align well with the ground truth.

**Weaknesses:**

I think this paper is not doing anything particularly novel. On the sampling side, the rates of Vempala and Wibisono are already somewhat outdated by the standards of the field and much better analysis is known for the LSI setting. See for instance some of the works on the proximal sampler.

Conversely, although I am less familiar with the learning theory elements, Theorem 3.2 appears to me to be straightforward. Thus, even if this exact result has not appeared in the literature before, the ideas in this paper do not seem particularly novel or surprising.

It is not really possible to assess whether this schema will be useful in practice, as the experiments appear somewhat small in scale. I would imagine that Gibbs sampling on large datasets is both expensive and unlikely to yield informative bounds.

**Questions:**

The terminology \emph{interpolation regime} is never defined.

219: iid -> i.i.d.\

224 has an extra comma

326:  the Theorem 1 -> Theorem 1

364: sand -> and

---

> ### Author Response · Authors · 2025-11-18
> **Official Response by Authors to Reviewer rD1H**
>
> We appreciate the reviewer's evaluation and comments. In what follows, we aim to respond to the reviewer's questions and additional comments.
>
> ### Weaknesses
> * **Novelty**: As acknowledged by the reviewer, we provide a rigorous data-dependent bound on the test error for overparameterized models in the interpolation regime. To the best of our knowledge, no prior work provides such a bound that is simultaneously data-dependent and applicable in the low-temperature regime. Since the reviewer complains about the lack of novelty, are you aware of comparable results in the literature?
>
> * **Regarding Theorem 3.2**: While we agree that the proof of Theorem 3.2 is straightforward, we would like to clarify that the key insight is identifying the correct functional $\Gamma$ for this setting and showing that it yields a tractable, informative generalization bound. We view the simplicity of the proof as a strength, as it allows the result to be broadly applicable.
>
> * **Sampling-related comments**: We also kindly ask the reviewer for a reference to more advanced sampling methods, as this might even further improve our results. Nevertheless, sampling is not really the principal focus of the paper.
>
> ### Questions and minor issues
> The interpolation regime is the scenario in which the model attains zero training error for any data.
>
> We have corrected the mentioned errors.

---

> > ### Author Response · Authors · 2025-11-28
> > **Follow-up on our rebuttal**
> >
> > As the discussion period is soon ending we would like to have feedback from the reviewer about whether we have answered their questions and have an opportunity to further engage in the discussion. We feel our contribution is significant because we present for the first time a tight data-dependent generalization bound for overparametrized neural networks, a problem of key importance in learning theory.

---

### Author Response · Authors · 2025-11-20
**Summary of Revisions**

We thank all reviewers for their constructive feedback. We have uploaded a revised version of the paper, where all modifications are highlighted in blue for ease of reference. Below we summarize the main changes introduced in response to the reviews.

### Summary of Revisions
* **Clarified the purpose and use of calibration**:
We explained why we need calibration to tighten the bound based solely on training data, while the bound itself also applies without calibration (as shown in Section C.2.6 ).
* **Added guidance on loss functions and applicability**:
We clarified the assumptions required on the loss and how we compute the bound (Section C.2.1), and expanded the experiments to include an unbounded loss function (Section C.2.7).
* **Improved presentation of theoretical components**:
Additional explanations were added to Corollary 4.2, addressing reviewer questions and increasing readability.
* **Corrected all identified typos and notational issues**:
All reported issues have been fixed. We thank the reviewers for their careful reading.

---

### Meta-Review · Area_Chair_DCJb · 2026-01-06

**Summary:**

The paper derives data-dependent generalization bounds for the Gibbs algorithm in an overparameterized interpolating regime, in which low training error is even observed for data with random labels. The bounds can be estimated via Langevin Monte Carlo approximations and are empirically validated on MNIST and CIFAR-10, where they are nontrivial for real labels and correctly upper-bound test error for random labels.

**Reviewer Concerns:**

Reviewers  main concerns were:
1. Missing novelty. The authors argued  that they are not aware of any other work that provides a rigorous data-dependent bound on the test error for overparameterized models in the interpolation regime.
2. Theorem 3.2 being straight forward.  The authors agreed and clarified that the main contribution lies in identifying the right data-dependent functional.
3. Practical estimation of the bound is computationally heavy and the estimation could be made more efficient by  using the posterior at higher temperatures as a kind of "warm-start" for lower temperatures. Authors agreed.

Unfortunately, all reviewers did not have high confidence and the reviews were quite short. Nevertheless, given the boarderline scores of the reviewers, I think the paper should not be accepted at this stage.

**Reviewer Scores:**

FGWz indicated that he will increase their score (from 2)

Reivwer V25C answered but did not indicate if they would increase or decrase their score. So I guess they will keep it.

Reviewer miYM submitted a blank review.

And the last (Rdh1) did not reply, but he had a score of 2, so I assume he would not sufficiently increase the score to turn this into acceptance.

---

### Decision · Program_Chairs · 2026-01-26

Reject